# Modeling of surface energy balance for Icelandic glaciers using remote sensing albedo

Andri Gunnarsson[1,3], Sigurdur M. Gardarsson[1], and Finnur Pálsson[2]

[1]University of Iceland, Civil and Environmental Engineering, Hjardarhagi 2-6, IS-107 Reykjavik, Iceland
[2]Institute of Earth Sciences, University of Iceland, Sturlugata 7, 101 Reykjavík, Iceland
[3]Landsvirkjun, Department of Research and Development, Reykjavík, IS-107, Iceland

**Correspondence:** Andri Gunnarsson (andrigun@lv.is)

**Abstract.** During the melt season, absorbed solar energy, modulated at the surface by albedo, is one of the main governing factors controlling surface-melt variability for glaciers in Iceland. An energy balance model was applied with the possibility to utilize spatio-temporal MODIS satellite-derived daily surface albedo driven by high-resolution climate forcing data to reconstruct the surface energy balance (SEB) for all Icelandic glaciers for the period 2000–2021. The SEB was reconstructed from April through September for 2000–2021 at a daily timestep with a 500 m spatial resolution. Validation was performed using observations from various glaciers spanning distinct locations and elevations with good visual and statistical agreement. The results show that spatio-temporal patterns for the melt season have high annual and inter-annual variability for Icelandic glaciers. The variability was influenced by high climate variability, deposition of light-absorbing particles (LAPs) from volcanic eruptions and dust hotspots in pro-glacial areas close to the glaciers. Impacts of LAPs can lead to significant melt enhancement due to lowering of albedo and increased short-wave radiative energy forced at the surface. Large impacts on the SEB were observed for years with high LAPs deposits, such as volcanic eruption years in 2004, 2010 and 2011 and the sand and dust-rich year of 2019. The impacts of volcanic eruptions and other LAP events were estimated using historical mean albedo under the same climatology forcing to provide estimations of melt energy enhancements. The impact of LAPs was often significant even though the glaciers were far away from the eruption location. On average, melt enhancements due to LAPs were ∼27% in 2010, ∼16% in 2011 and ∼14% in 2019, for Vatnajökull, Hofsjökull and Langjökull.

*Copyright statement.* TEXT

## 1 Introduction

Mass and energy balance changes of glaciers are useful indicators of changes in the cryosphere and climate (e.g., Jóhannesson, 1986; Jóhannesson et al., 1989; Slater et al., 2021). Projected future climate changes in the Northern Hemisphere would force reduction in the area and volume of existing glaciers and ice sheets, contributing significantly to global sea level rise (e.g., Gregory and Oerlemans, 1998; Zemp et al., 2019; Schmidt et al., 2020; Hofer et al., 2020; Goelzer et al., 2020). In the Northern Hemisphere absorbed short-wave energy during the melt season is the primary energy source for surface melting of

snow and glaciers (e.g., Male and Granger, 1981; Björnsson and Pálsson, 2008; Fernandes et al., 2009; Hudson, 2011; Box et al., 2012; Chen et al., 2016). Albedo of snow- and ice-covered surfaces is the unitless ratio of the radiant flux reflected from Earth's surface to the incident flux and thus accurate representation of albedo is critical to understand and model surface melt (Schmidt et al., 2017). Changes in snow- and ice-cover duration and extent, can magnify the effect on climate for warming and cooling due to the complex and self-enhancing ice-albedo feedback with temperature (Barnett et al., 2005; Adam et al., 2008; Choi et al., 2010; Hudson, 2011; Flanner et al., 2011; Box et al., 2012). Given the importance of snow- and ice-albedo as an amplifier of climate change, surface albedo has been defined as an Essential Climate Variable and a requirement for climate monitoring (WMO, 2011; Bojinski et al., 2014).

Iceland is an island (103,000 km$^2$) located at major climatic boundaries in the North Atlantic Ocean, where changes in atmospheric circulation and ocean currents influence the climate. Iceland has a maritime climate with mild winters, cool summers and high average precipitation, especially in the fall and winter, sustaining a seasonal snow pack and glaciers (Einarsson, 1984; Perkins et al., 1998). The North Atlantic Current, a northeastward-flowing branch of the Gulf Stream, transports warm ocean water to the North Atlantic Subpolar Gyre, explaining milder climates at higher latitudes (Lozier et al., 1995; Rossby, 1996; Ólafsdóttir et al., 2010; Knudsen et al., 2012). Flowing along the southern and western Icelandic coast the Irminger Current brings relatively warm Atlantic water towards Iceland, moderating the climate. The cold East Greenland Current, originating in cold Polar waters, and the cold East Icelandic Current, a branch of the East Greenland Current, bring cold water masses towards the Icelandic coast in the north and east, respectively (Renner et al., 2018; Zhao et al., 2018). Associated with anthropogenic warming, global sea surface temperatures (SSTs) have been observed to increase in the past century. Future projections indicate further warming, although increased melting of the Greenland ice sheet and Arctic sea ice has been linked with local SST cooling south of Greenland, a region referred to as the North Atlantic warming hole (NAWH), with possible impacts on the surface mass balance of Icelandic glaciers (Rahmstorf et al., 2015; Alexander et al., 2018; Gervais et al., 2019; Keil et al., 2020; Noël et al., 2022).

The total area of glaciers in Iceland in 2019 was approximately 10,400 km$^2$ ($\sim$10 % of Iceland), containing about 3400 km$^3$ of ice (in 2019), corresponding to $\sim$9 mm of potential global sea level rise (Björnsson and Pálsson, 2020; Aðalgeirsdóttir et al., 2020; Hannesdóttir et al., 2020). The mass balance of Icelandic glaciers has changed significantly over the last three decades, and all studies and projections indicate that the mass loss of Icelandic glaciers will continue and increase with accelerated warming in the Northern Hemisphere in the future (Jóhannesson et al., 2007; Schmidt et al., 2020; Noël et al., 2022).

Iceland has about 22,000 km$^2$ of sandy deserts that are a major source of atmospheric dust and light-absorbing particles (LAPs) (Arnalds et al., 2016). Many of those areas are near glaciers and are sources of active dust emission, defined as dust hotspots (e.g., glacio-fluvial plains, sand plains), with unstable surfaces and are prone to dust aerosol production that can deposit in snow and glacier surfaces, influencing the surface albedo and thus the radiative forcing (Björnsson and Pálsson, 2008; Wittmann et al., 2017; Dagsson-Waldhauserova et al., 2017; Gunnarsson et al., 2021).

Glacier mass and energy balance models generally do not simulate albedo changes caused by atmospheric dust and LAP deposition, as the processes involved are complex to model and dust sources can be far away from the glacier surface. In volcanic regions, eruptions can produce vast amounts of volcanic ash of diverse grain size and even extremely thin tephra

deposits on snow- and ice surfaces can lead to significantly enhanced melt potential; but in cases of a thick tephra layer deposits (>∼2 cm) prevent surface melt processes (Warren and Wiscombe, 1980; Möller et al., 2014; Wittmann et al., 2017; Möller et al., 2019; Gunnarsson et al., 2021). The majority of mass loss from glaciers in Iceland is due to surface mass balance processes. However, non-surface mass balance is non-negligible (through the processes of geothermal activity, volcanic eruptions, geothermal heat flux, calving, internal friction and water flow) even though these processes amount to only a fraction of the surface ablation (Björnsson et al., 2001, 2013; Aðalgeirsdóttir et al., 2020; Jóhannesson et al., 2020).

Research and monitoring of Icelandic glaciers is important for a range of reasons, e.g., civil security (because of jökulhlaups), sub-glacial volcanic activity, stability of river paths, runoff variability, long- and short-term changes due to climate change, natural variability, and water resource forecasting for efficient hydro-power production. Efficient water resource utilization requires forecasting on sub-daily, daily and seasonal timescales for operational planning. Longer timescales (years and decades) are also important for refurbishment of older hydro power infrastructure as part of climate change adaption and development of new hydro power plants (Jóhannesson et al., 2007; Sveinsson, 2016). Hydro-power production accounts for about 70 % of total energy production in Iceland. In an average hydrological year, about 50 % of inflow to reservoirs and diversions for hydro power energy production originates from annual glacier melt (Hjaltason et al., 2020). Additionally, because the current Icelandic energy system is a closed loop system, with no import or export of energy (except fossil fuel), high-quality forecasting capabilities are desirable.

The primary objective of this study is to understand and quantify melt season SEB for Icelandic glaciers using high-resolution meteorological climate forcing and remotely sensed glacier surface albedo from the Moderate Resolution Imaging Spectrora-diometer (MODIS) sensor. The study builds on processing pipelines for albedo developed in Gunnarsson et al. (2021). This adds to the previous understanding of spatial and temporal distributions of melt energy, main melt energy sources, and variability within and between glaciers in Iceland and provides insight into the melt enhancement due to volcanic eruptions and years with extensive LAP deposits. In the case of future volcanic eruptions or extensive LAP events, the presented methodology allows for rapid assessment of glacier albedo changes in near-real time and the associated influence on surface energy balance, which can have a direct impact on glacier runoff and thus on hydropower production in Iceland and possibly civil infrastructure in some cases. Understanding LAPs processes and impacts on SEB also aids in parametrizations of albedo for other modelling work where remotely sensed albedo may not be available, such as for historic and future modelling studies. The study also provides a comprehensive overview of the SEB of Icelandic glaciers since it is not limited to one glacier or glacier outlet as many previous studies of surface energy balance.

## 2  Study area

The analysis in this study covers the six largest Icelandic glaciers, namely Vatnajökull, Langjökull, Hofsjökull, Drangajökull, Mýrdalsjökull and Eyjafjallajökull (altogether about 97 % of the glaciated area in Iceland), although the model described in the study was applied to all the glaciers in Iceland. Figure 1 shows the outlines (black) of the six glaciers, and their division into main ice flow basins for detailed analysis. The ice flow basins are named according to the first letter of the respective ice

cap and their location (e.g., VNW for the northwestern outlet of Vatnajökull). Catchment delineation is from Magnússon et al. (2016a) for Drangajökull; Björnsson (1988) and Björnsson et al. (2000b) for Hofsjökull and Mýrdalsjökull; and Pálsson and Gunnarsson (2015); Pálsson et al. (2013); Pálsson et al. (2016) for Langjökull and Vatnajökull. The sub-areas are chosen as in Gunnarsson et al. (2021). For the six glaciers and the defined ice flow basins, topographic properties were extracted: area, mean, maximum and minimum elevation, as shown in Table 1. Figure 1 also shows the locations of ICE-GAWS sites, used for validation purposes in this study, with grey dots. Glacier outlines were kept fixed throughout the study period (2000–2021) using the available delineation spanning 2007–2013 from Hannesdóttir et al. (2020). Since annual data was not available, this was selected as a midpoint representing an average glacier extent during the study period. Care must be taken when interpreting the results at glacier terminus areas, as active glaciated areas in 2000 might be dead ice or land in more recent time.

Over the study period, three volcanic eruptions were observed in glaciated areas with extensive LAP deposits, affecting the SEB. In November 2004, an eruption in the sub-glacial volcano Grímsvötn, lasting 5-6 days, produced an estimated $\sim$0.06 km$^3$ bulk volume of tephra (Jude-Eton et al., 2012; Oddsson et al., 2012). The tephra deposits from the eruption were mainly distributed northeast of Grímsvötn in a narrow plume (Oddsson et al., 2012). In 2010, an eruption in Eyjafjallajökull started on April 14 and lasted 23 days. The tephra plume, carrying an estimated volume of $\sim$0.27 km$^3$ of tephra, was mostly directed towards the south and southeast (Gudmundsson et al., 2012). During the last days of the eruption, a short period of diverse wind directions brought notable LAP deposits to all the major glaciers. A second eruption occurred in Grímsvötn on May 21, 2011, lasting 7 days and releasing an estimated $\sim$0.8 km$^3$ of basaltic tephra (Hreinsdóttir et al., 2014). The tephra deposits were mostly distributed south and southwest during the eruption, but a thin layer was noticeable on all of western Vatnajökull, and regions in the southeast (Hreinsdóttir et al., 2014).

## 3 Data and methods

### 3.1 Meteorological in situ data

The Icelandic Glacier Automatic Weather Stations network (ICE-GAWS) stores meteorological observations from AWSs located at sites on Vatnajökull, Langjökull, Mýrdalsjökull and Hofsjökull since 1994, 2001, 2015 and 2016, respectively. Most of the stations were operated during the ablation season, from May through September annually, but a few operate all year round. In total, 20 sites provided data for the study period; all the sites measure air temperature and incoming short-wave radiation, while 13 sites also measure incoming long-wave radiation. Details on data processing are provided in Gunnarsson et al. (2021). Table A1 provides details of the location, elevation and number of observations for each site in the Appendix, and Figure 1 shows their location. In the current study, observations of air temperature, short- and long-wave incoming radiation were used for validation purposes.

## 3.2 Surface albedo and cloud cover

For the SEB model applied in this study, snow- and ice-surface albedo ($\alpha$) were derived from MODIS data using processing models developed for Iceland by Gunnarsson et al. (2021). The products rely on the MOD10A1 (Terra satellite) and MYD10A1 (Aqua satellite) snow albedo (Scientific Data set: Snow Albedo Daily Tile) for the grid tile h17v02, covering most of Iceland except for a small portion of the Snæfellsnes peninsula. Data were collected from the US National Snow and Ice Data Center (NSIDC) (Hall and Riggs, 2016a, b) for further processing. For the period from February 23, 2000 to May 4, 2002, albedo data were only based on Terra since Aqua was not yet in orbit. A total of 62 dates were missing for MOD10A1, and 12 for MYD10A1, for the study period (April through September each year), excluding data missing due to polar darkness from late November until late January each year. The albedo data produced in Gunnarsson et al. (2021) was based on version 6.0 of the MODIS data but in this study reprocessed using version 6.1 without modifications to the processing steps.

For snow- and ice-surface albedo, daily merging was applied to the Terra MOD10A1 and Aqua MYD10A1 albedo data to reduce the number of cloud-obscured daily pixels, i.e., all non–cloud obscured pixels were merged into a single tile from both products daily. Temporal aggregation was then applied to further reduce the number of cloud-obscured pixels. The temporal aggregation range was set as the number of days backwards and forwards from each center date to merge into a single stack for further processing. A temporal aggregation range of 5 days backward/forward was selected, allowing 11 days from both MOD10A1 and MYD10A1 to contribute data to the temporally aggregated product. This results in a total of 22 values that are potentially available for each pixel (i.e., 11 days of MOD10A1 and 11 days of MYD10A1). For each data stack, containing the potential 22 values contributing albedo data, the mean was calculated to represent the daily surface albedo, after median-based statistical rejection of outliers. The remaining pixels classified as clouds were classified statistically with four predicting variables, location (easting, northing), elevation (Z), and aspect, with a daily-trained random forest model. Further information and details are in Gunnarsson et al. (2021).

Cloud cover data were based on the classifications of clouds in the M*D10A1 products (M*D35_L2 cloud mask). For each day the two tiles were merged, creating a daily cloud cover estimate at the time of satellite overpass (10:30 AM and 1:30 PM local time), i.e., cloud or no cloud. Data were then aggregated to monthly and melt season mean values accordingly.

## 3.3 Model forcing

Meteorological forcing based on the Weather Research and Forecast model (WRF version 3.6.1) coupled to the NOAH land surface model was used to provide climatological surface variables at a 2 km spatial resolution and 1 hour temporal resolution spanning the study period from January 1, 2000, to September 30, 2021. For the bulk of the period, data from the WRF-RAV2 configuration were used, developed for the climatological reanalysis RAV2 project (RAV2) (Rögnvaldsson, 2016). Since RAV2 was forced with boundary conditions from the European Centre for Medium Range Weather Forecasts (ECMWF) ERA-Interim reanalysis (Berrisford et al., 2011), data availability, overlapping the current study, spans from January 1, 2000 out through August 2019 due to the end-of-life of ERA-Interim.

To extend the range of data availability past August 2019, the climatological reanalysis was extended with model configuration nearly identical to RAV2, but outer domain boundary conditions forcing was from the Global Forecast System (GFS) from the US National Centers for Environmental Prediction (NCEP) weather forecast model (National Centers for Environmental Prediction, 2015). This product is referred to as the IceBox model configuration (IceB) and provides data from September 1, 2018 to September 1, 2020. The IceBox model configuration was also run operationally, as a forecasting system, providing data four times per day, referred to as FCST. To extend the analysis range even further, allowing the energy balance model presented here to cover the extraordinary summer weather in 2021, the forecast data from the IceBox domain starting at April 1, 2020 was aggregated to an hourly time series using the shortest forecasting step in each case. Further description of the RAV2 and IceBox model setup and output configuration is found in Rögnvaldsson (2016, 2020). Relevant meteorological surface data were extracted for use in the energy balance model, including air temperature at 2 m, wind speed, incoming long- and short-wave radiation, barometric pressure at surface level, and specific humidity; all were resampled to daily average values.

To downscale the meteorological forcing data from the 2 km WRF grid to the 463 m MODIS grid, the model uses the IslandsDEM digital elevation model from the National Land Survey of Iceland (accessed June 1, 2020, https://atlas.lmi.is/mapview/?application=DEM). A 20 m version of the elevation model was resampled to the native MODIS grid for further processing with bi-cubic interpolation using the *griddata* function in Matlab (Matlab, 2020). Elevation-dependent variables (air temperature and long-wave radiation) were adjusted for the difference between the coarse-resolution WRF DEM and high-resolution IslandsDEM at 463 m using lapse rates. Other meteorological forcing data was downscaled with bi-cubic interpolation.

Statistical downscaling of temperature requires a temperature lapse rate, often taken to be the free-air moist adiabatic lapse rate ranging from 6–7 K km$^{-1}$ (Stone and Carlson, 1979). Hodgkins et al. (2013) investigated temperature lapse rates for outlet glaciers at Langjökull and southeast Vatnajökull during 2003–2007. They reported mean monthly lapse rates ranging from 4.5 K km$^{-1}$–8.0 K km$^{-1}$, with clear monthly variations from April to October. Generally a higher lapse rate (6.5–8 K km$^{-1}$ (mean for April, May and September as 7.0 K km$^{-1}$)) was seen in spring and fall, with lower rates in summer (mean for June, July, August and September as 5.7 K km$^{-1}$). This is in good agreement with results from Gardner et al. (2009) for Arctic glaciers, with an ablation season mean of 4.9 K km$^{-1}$. Crochet and Jóhannesson (2011) developed a one-parameter terrain model with a constant vertical lapse rate of 6.5 K km$^{-1}$ with temperature observations for Iceland, excluding glaciers, validated to ground base. Their results suggested that the assumption of a 6.5 K km$^{-1}$ lapse rate was applicable in Iceland. Further work by Nawri et al. (2012) supports this. Here for glaciated areas a temperature lapse rate of 7.0 K km$^{-1}$ was applied for JFMA (January–April) and SOND (September–December) while 5.5 K km$^{-1}$ was applied for the active melt season, MJJA (May–August), following the results from Hodgkins et al. (2013).

Downward long-wave radiation is primarily determined by humidity and temperature vertical atmospheric profiles and thus is a function of elevation (Plüss and Ohmura, 1997; Ohmura, 2001). Hinkelman et al. (2015) used a constant long-wave radiation gradient of 29 W m$^{-2}$ km$^{-1}$ to correct for varying elevation (Marty et al., 2002). Enhancement of long-wave radiation by surrounding terrain emission is important when sky radiation is low, e.g., in cold and dry atmospheres, generally at high elevations with steep topography (Sicart et al., 2006). In this study no adjustments were made to account for enhancement of

long-wave radiation due to terrain emission, as the effect is small on the large concave ice caps investigated. A lapse rate of 29 W m$^{-2}$ km$^{-1}$ was used for elevation difference adjustment between the WRF and MODIS grids.

In many studies, incoming short-wave radiance is separated into beam and diffuse components and corrections are made for terrain elevation, slope and tree-cover fractions (Bair et al., 2016; Rittger et al., 2016). Here, since the WRF data only provide total incoming short-wave radiation, no adjustments were made in this regard.

### 3.4 Surface energy balance

The physical processes driving surface melt over snow- and ice-covered surfaces are isolated using estimations of the SEB at a daily timestep with a spatial resolution of 463 m. The SEB model used in this study was adopted from that of van As et al. (2005), which has previously been used on weather station data on the Greenland ice sheet (Van As, 2011; Charalampidis et al., 2015; van As et al., 2017; Vandecrux et al., 2018). The model was modified for the input model forcings, described in the previous sections.

The SEB was closed by iteratively solving for surface temperature ($T_s$):

$$SW \downarrow (1-\alpha) + LW \downarrow + LW \uparrow (T_s) + SHF(T_s) + LHF(T_s) + G + M = 0 \tag{1}$$

where $SW \downarrow$ is the incoming short-wave radiation, $\alpha$ is broadband albedo, $LW \downarrow$ and $LW \uparrow$ are the incoming and outgoing long-wave radiation, respectively, $SHF$ is the sensible heat flux, $LHF$ the latent heat flux and $G$ the sub-surface heat flux (assumed zero), with the fluxes defined positive when adding energy to the surface. $M$ is the energy surplus used for surface melt. Solutions for $T_s >$ 273.15 K indicate availability of melt energy. If $T_s$ was > 273.15 K, $T_s$ was set as 273.15 K and melt $M$ was computed, otherwise, if $T_s$ was $\leq$ 273.15 K, $T_s$ was set to 273.15 K (0°C) and no melt assumed.

Outgoing long-wave radiation (LW↑) defines the energy emitted to space by Earth's surface and depends on surface temperature. Here, outgoing long-wave radiation was calculated based on the Stefan–Boltzmann law:

$$LW \uparrow = \varepsilon \sigma T_s^4 \tag{2}$$

where $\varepsilon$ is the broadband emissivity of snow and ice (0.98) (Salisbury et al., 1994) and $\sigma$ is the Stefan–Boltzmann constant.

Turbulent fluxes of sensible heat $SHF$ and latent heat $LHF$ were estimated using the bulk aerodynamic approach with stability corrections based on Monin–Obukhov similarity theory (van As et al., 2005; Smeets and van den Broeke, 2008a). The sensible (SHF) and latent (LHF) heat fluxes are expressed as:

$$SHF = \rho c_p u_* T_* \tag{3}$$

$$LHF = \rho \lambda u_* q_* \tag{4}$$

where $\rho$ denotes air density; $c_p$ is the specific heat of dry air at constant pressure (1005 J K$^{-1}$ kg$^{-1}$); $\lambda$ is the latent heat of sublimation; $u_*$ is the friction velocity; $T_*$ and $q_*$ are turbulent scales of temperature and humidity, respectively, defined as:

$$u_* = \frac{\kappa u(z)}{\ln(z/z_0) - \psi_m(\xi)} \tag{5}$$

$$T_* = \frac{\kappa(T(z) - T(0))}{\ln(z/z_T) - \psi_T(\xi)} \tag{6}$$

$$q_* = \frac{\kappa(q(z) - q(0))}{\ln(z/z_q) - \psi_q(\xi)} \tag{7}$$

where $\kappa = 0.4$ is the von Kárman constant; $u$, $T$ and $q$ are wind speed, air temperature and humidity at height $z$ and $z_0$,
$z_T$, $z_q$ are surface roughness lengths associated with these parameters. The stability correction functions for momentum ($\psi_m$),
heat ($\psi_T$) and humidity ($\psi_q$) depend on the stability parameter $\xi =$ z/$L_*$ where $L_*$ is the Obukhov length scale. The stability
functions of Holtslag and Bruin (1988) for stable stratification and Paulson (1970) for unstable stratification are used.

Surface roughness lengths for heat and moisture were calculated for snow and ice separately as in Van As (2011). The surface
roughness length for momentum ($z_0$) varies considerably in time and space and generally is set to different constant values for
snow- and ice surfaces (Brock et al., 2006; Smeets and van den Broeke, 2008b). Reported values for surface roughness lengths
of momentum range from 1 to 10 mm while lower values generally apply for snow (0.1 mm) (Brock et al., 2006). Values up to
60 mm have been reported at Breiðamerkurjökull where ice hummocks up to almost 2 m in height can be formed during the
melt season but are not representative for the majority of bare-ice areas of glaciers in Iceland (Smeets et al., 1999; Wildt et al.,
2004).

Guðmundsson et al. (2009) applied $z_0$ as 0.1, 2 and 10 mm for new snow, melting snow and ice in the ablation zone
respectively, in a SEB model for Langjökull, and Wildt et al. (2004) used similar values for Vatnajökull. Schmidt et al. (2017)
applied a constant value of 1 mm for both snow and ice when modeling the energy balance for Vatnajökull. Since no data
exist on spatio-temporal variability of $z_0$ for glaciers in Iceland, a simple classification scheme discriminating between snow
and bare ice was applied based on surface albedo. For pixels with albedo values lower than or equal to 0.45 (bare ice), $z_0$ was
assigned as 3 mm; for pixels with albedo higher than 0.45, $z_0$ was assigned as 1 mm (snow).

Potential melt water was defined as the direct conversion of melt energy into water equivalent using latent heat of fusion
(0.26 mm day$^{-1}$ per W m$^{-2}$ ).

## 4   Results and discussion

### 4.1   Validation of meteorological forcings and model outputs

The downscaled meteorological forcings used for calculations of daily incoming short-wave radiation (SW↓), incoming long-
wave radiation (LW↓), air temperature, outgoing short-wave radiation (SW↑), outgoing long-wave radiation (LW↑) and relative
humidity (RH) from WRF were validated with in situ data. Figure 2 shows a comparison of observed and simulated daily air
temperature at 2 m height, SW↓ and LW↓ for the different WRF model configurations, RAV2, ICEB and FCST. Generally,
for the whole validation period, from April 1 to October 30 each year, the results show good agreement, both visually and
statistically, for all configurations and are within ranges reported by Schmidt et al. (2017). Table 2 shows the validation results

for the whole validation period, similar to Figure 2 but also for each month within the full period. Table A2 in the appendix shows statistical validation results for SW↑, LW↑ and relative humidity.

For air temperature, $R^2$ is 0.83, 0.93 and 0.94 for the RAV2, ICEB and FCST configurations, respectively, for the whole period from April through October. For all configurations, for both the full period and monthly intervals, the temperature bias was negative in the range of -0.27 to -1.15 K. The smallest bias values were observed in July and August, with slightly higher values closer to spring. The consistent negative bias indicates that the model slightly overestimates air temperature.

Daily average SW↓ RMSE ranged from 24 to 62 W m$^{-2}$, with the highest values during summer coinciding with the summer solstice. $R^2$ is 0.63, 0.67 and 0.62 for the RAV2, ICEB and FCST configurations, respectively, for the whole period from April through October. For RAV2 the bias was mostly positive, ranging from 9 to 25 W m$^{-2}$, with the exception of September and October which have slightly negative bias values. The lower and negative values might be related to larger solar zenith angles as less incoming short-wave energy was available. During these months the contribution to melt from short-wave radiation was generally limited. For both ICEB and FCST, RMSE values were similar to results for RAV2 but bias values were more consistently negative. In this comparison, far fewer sites were available for validation because of the limited temporal range of the ICEB and FCST configurations.

LW↓ agreement was good, with RMSE from 9 to 16 W m$^{-2}$, $R^2$ ranging from 0.44 to 0.86 and a general negative bias from -2 to -18 W m$^{-2}$, with the exception of a mean bias of -16 W m$^{-2}$ in April for RAV2. These outlying values might be related to the fact that spring maintenance of the ICE-GAWS stations generally takes place in late April or early May. The mean bias was consistently highest in April for all WRF configurations and generally decreases into the summer months. The instrument-reported uncertainty in daily totals was less than 5 % ($\sim$15 W m$^{-2}$) for short-wave radiation and less than 10 % ($\sim$30 W m$^{-2}$) for long-wave radiation, which could partly explain some of the discrepancies.

Daily average relative humidity (RH) had RMSE ranging from 2 to 6 % for all the WRF configurations, $R^2$ ranged from 0.38 to 0.95, with an average value of 0.65. Bias values ranged from 4 to 11 %. Limited temporal patterns were observed between periods. Mean observed relative humidity from the automatic weather station network (AMJJASO) was 85.8 % with a standard deviation of 15 % indicating limited variability. LW↑ RMSE values were similar to LW↓, overall a bit lower, ranging from 2 to 9 W m$^{-2}$. LW↑ $R^2$ were lower than for LW↓. Since LW↑ is constant of $\sim$316 W m$^{-2}$ under melting conditions (surface temperature = 273.15 K), average observed LW↓ was 306.58 W m$^{-2}$ while 304.81 W m$^{-2}$ for the model, over the active melt season AMJJASO. SW↑, estimated thorough the WRF meteorological forcings and MODIS remotely sensed albedo, had an average RMSE value of 36 W m$^{-2}$, up to 70 W m$^{-2}$, average $R^2$ was 0.47 and a negative bias of -14 W m$^{-2}$. Further statistical details are in Table A2.

Overall the performance of the different WRF configurations was similar, although it should be noted that the data period for RAV2 data spans 19 years while far fewer data were available for validation of the ICEB (2 years) and FCST (2 years) configurations. Individual station comparison reveals no prominent patterns related to station elevation or location. Recent work by Schmidt et al. (2017) reported similar results when validating HIRHAM5 for surface mass balance calculations for Vatnajökull, while recent work by Huai et al. (2020) validating ERA-Interim and ERA5 against the PROMICE weather station network on the Greenland ice sheet reports overall better comparison for the same statistical parameters. One explanation of the

difference might relate to the lower overall cloud cover over the Greenland ice sheet, compared to glaciers in Iceland, impacting weather simulations. Another explanation might relate to the fact that PROMICE short-wave radiation data are post-processed to adjust for station tilting, as inaccurate measurements in clear-sky conditions are expected, giving rise to better comparison (Van As, 2011; Fausto et al., 2021). Validation of MODIS albedo was done in Gunnarsson et al. (2021) and Gascoin et al. (2017) for glaciers in Iceland.

## 4.2 Surface energy balance seasonal and inter-annual variability

Figure 3 shows the spatial patterns for melt energy for the investigated glaciers over the period 2000–2021 for individual months in the extended melt season (AMJJA) and the extended melt season mean (AMJJA). Spatially, the highest melt energy was observed where the winter snow cover is generally completely ablated during summer, revealing dirty and impurity-rich ice. Lower melt energy values were found in the accumulation areas associated with higher elevations and a shorter period of positive SEB during the melt season. In April, limited melt occurs, although in areas near the terminus at the north and south Vatnajökull outlets and low-lying outlets of Mýrdalsjökull, between 10 and 15 % of the total mean annual melt energy was observed. At the northern outlets of Vatnajökull, winter snow thickness is generally shallower than for other outlets, exposing impurity-rich ice with low albedo sooner and enabling greater amounts of the incoming short-wave radiation to be forced at the surface. At the lower elevations of Vatnajökull southern outlets, some extending down to sea level, average winter and spring temperatures are higher, inducing earlier melt-out of winter snow, which exposes impurity-rich ice and portions of the ablation area in April. In spring and early summer, the positive SEB contributes to the warming and ripening phase of the winter snow before the melt output phase contributing to melt can commence. The highest daily amounts of incoming short-wave energy occur in June and July, providing the largest quantities of melt energy associated with small solar zenith angles. As more impurity-rich ice was exposed in the ablation area, with lower surface albedo as the melt season progresses, more incoming short-wave energy was available at the surface, even in August with increasing solar zenith angles. Gunnarsson et al. (2021) revealed that the lowest observed albedo values in the accumulation area generally occur in early to mid-August prior to precipitation falling as snow, and thus higher albedo, reducing short-wave net radiation ($SW_{net}$).

Figure 4 shows the average SEB (MJJA) and its main components as a function of elevation in 100 m bands for the six largest ice caps and their sub-areas defined in Figure 1. For all the glaciers, $SW_{net}$ was the major SEB component for melt energy while $LW_{net}$ was generally an energy sink. The sensible heat flux (SHF) was an energy source in the lower ablation area, generally decreasing with elevation as air temperatures decrease. High air temperatures, explained by the Icelandic maritime climate during summer, and generally low glacier elevations, explain the positive melt energy contribution of SHF. Latent heat flux (LHF) was quite small in all cases, with much less variability with elevation than other melt energy components. Due to high humidity, the LHF was mostly positive.

For Vatnajökull, $SW_{net}$ diminishes on average -6.45 W m$^{-2}$ per 100 m with lower gradients for the northeastern and northwestern outlets (-8.95 and -11.1 W m$^{-2}$ per 100 m, respectively). For Hofsjökull lower values were observed for the southwestern outlets (-9.0 W m$^{-2}$ per 100 m), with -6.8 to 8.0 W m$^{-2}$ per 100 m for the southeastern and northern outlets, respectively. At Langjökull the northeastern and northwestern outlets have lower gradients (-9.2 and -10.1 W m$^{-2}$ per 100 m)

than the southern outlet (-8.7 W m$^{-2}$ per 100 m). The smaller glaciers had similar average values, -8.2, -5.9 and -6.0 W m$^{-2}$ per 100 m for Eyjafjallajökull, Mýrdalsjökull and Drangajökull, respectively.

Compared to other glaciers and ice sheets studied in the Northern Hemisphere, in the European Alps, Greenland and Svalbard, the results obtained are similar. SW$_{net}$ is generally the main energy source for heating and melting of snow and ice during the melt season, and net long-wave radiation is an energy sink with a significant contribution from sensible heat fluxes, although significant variations can be found (Sicart et al., 2008; Oerlemans et al., 2009; van den Broeke et al., 2011; Franco et al., 2013; Karner et al., 2013; Huai et al., 2020). Partitioning of the SEB reveals a somewhat higher contribution from SW$_{net}$ for Icelandic glaciers than other Northern Hemisphere glaciers and ice sheets, although dependent on local glacier conditions (Hock, 2005; Six et al., 2009). This was driven by lower albedo values due to LAPs, both deposited in the surfaces of glaciers during summer and also from historical eruptions and dust events melting out in the impurity-rich bare-ice areas during the ablation season. Generally, net radiation contribution by LW$_{net}$ was mostly an energy sink (negative), reducing the SW$_{net}$ contribution, increasing the relative contribution of the sensible heat fluxes to melt.

Latent heat fluxes contribute much less than sensible heat fluxes. Variation of SW$_{net}$ with elevation depends strongly on albedo, generally increasing with elevation, as impurity-rich ice was exposed later in the melt season, or not at all, in the accumulation area. General albedo evolution in the accumulation area throughout the melt season was mainly driven by climatology, i.e., snow metamorphosis, not LAPs, although events of sand- and dust deposits could be observed in the albedo data for individual years, impacting SW$_{net}$. MODIS albedo data does not allow for discrimination between snow metamorphosis and LAPs impacts, but this assumption was based on albedo data in the accumulation area that seldom reach values low enough to reflect extensive LAPs in the surface, unless related to years with volcanic eruptions. Figure 6 in Gunnarsson et al. (2021) shows the average elevation distribution of albedo.

Albedo gradients from Gunnarsson et al. (2021) follow similar patterns with elevation to those of SW$_{net}$ (general albedo increase with elevation) for all the glaciers, demonstrating how SW$_{net}$ was modulated by albedo. Cloud cover also influenced SW$_{in}$, generally increasing slightly with elevation, although persistent cloud cover was observed at the terminus at Vatnajökull northern outlets. Spatial distribution of cloud cover has been reported in Gascoin et al. (2017) and Gunnarsson et al. (2021). LW$_{net}$ was negative (energy sink) for all the glaciers, and generally more negative with greater elevation. The variability was much less than for SW$_{net}$, ranging from -1.8 to - 0.2 W m$^{-2}$ per 100 m. Changes to SHF with elevation are similar to those of LW$_{net}$, reducing by - 1.62 to 0 W m$^{-2}$ per 100 m, with limited elevation variability for Mýrdalsjökull and especially Drangajökull. LHF fluxes were small in all cases, with non-significant elevation dependency.

Figure 5 shows the variation of average monthly melt energy anomaly and albedo anomalies in 100 m elevation bins for Vatnajökull, spanning individual months from April through September for the study period. The anomalies show deviations from the period mean for each month and elevation bin. In years with high summer ablation, increased melt energy in the accumulation area was observed. The bare-ice areas generally reach a certain lower limit of albedo (0.1–0.25), limiting further effects of albedo on short-wave radiative forcing, although the timing of bare-ice exposure is important. Figures B1 to B5 in the Appendix show similar patterns for the other glaciers studied.

The figure shows that in 2010 and 2011, tephra deposits in the upper elevations, from the eruptions in Eyjafjallajökull (2010) and Grímsvötn (2011), greatly impacting albedo in the accumulation area. In 2012, below-average cloud cover extensively enhanced $SW_{in}$ radiation forcing, while some residual effects from tephra fallout in 2010 and 2011 were possible, increasing $SW_{net}$. Positive melt energy anomalies at lower elevations in 2015, 2016 and 2017 were related to rapid lowering of albedo associated with warm southerly winds and liquid precipitation in the first months of the melt season. Much colder temperatures and cloudy periods followed, constraining melt energy during the rest of the melt season. The high SEB in 2019 was largely due to negative albedo anomalies, resulting from extensive LAP deposits from the near pro-glacial areas (unpublished data, based on satellite images and operational web cameras in the field). This extended the actively melting areas higher into the usual accumulation zone, contributing more to the summer ablation by increasing melt at higher elevations. The year 2021 was unusual, as May and the first three weeks of June were highly influenced by clear skies but cold temperatures, the latter reversing completely in late June, with warm westerly and southerly winds and clear skies through August (Pálsson et al., 2022; Veðurstofa Íslands, 2022). Figure 6 shows the SEB for the study period and the decomposition into different SEB components, with melt season mean cloud cover and albedo anomalies. The SEB variability between melt seasons is mostly explained by $SW_{net}$ variability while $LW_{net}$ and SHF partially explain the variance. As shown in Figure 6, the latent heat flux made only a limited contribution to melt energy. $SW_{net}$ was the dominant melt energy source for all locations studied. $LW_{net}$ acts as an energy sink, ranging from -20 to -30 W m$^{-2}$, with variability between the glaciers investigated around 4 W m$^{-2}$. The figure shows that for Vatnajökull the melt season average SEB components were 97 ($\sigma$: 14.5 ), -30 ($\sigma$: 4.2 ), 16.6 ($\sigma$: 2.1 ) and 2.7 ($\sigma$: 1.1 ) W m$^{-2}$ for $SW_{net}$, $LW_{net}$ , SHF and LHF for the period, respectively. Excluding 2010 and 2011, the $SW_{net}$ was 92 ($\sigma$: 10.5 ) due to the enhancement effects of the volcanic eruptions for those years, and the average energy available for melt was 85 W m$^{-2}$ for Vatnajökull. Higher $SW_{net}$ was observed at the south-coast glaciers (Mýrdalsjökull, Eyjafjallajökull), which tend to have very low albedo values and earlier melt onset in spring. The south-coast glaciers were also close to unstable dust hotspot areas where seasonal snow melts out earlier than in the highlands, exposing erosive surfaces. Conversely, cloud cover was generally higher for the south-coast glaciers, as well as at the coastal Drangajökull in the northwest, with cloud cover ranging from 75 to 82 %, while less cloud cover, 70–74 %, was observed for the inland glaciers and their outlets, Vatnajökull (except SE outlets), Langjökull and Hofsjökull. $SW_{net}$ was strongly affected by both cloud cover and surface albedo; lower albedo and cloud cover values were observed for areas of high annual melt energy. $SW_{net}$ correlates strongly with the average surface albedo (Pearson Correlation Coefficient, PCC: - 0.85), where a general increase in albedo with a consequent decrease in $SW_{net}$ was reduced with longitude. (Gunnarsson et al., 2021). However, a non-significant relationship was found between cloud cover and $LW_{net}$ (PCC: 0.72 for Vatnajökull).

Figure 7 shows the monthly average distribution of SEB components and melt energy for the glaciers studied. For nearly all glaciers the $SW_{net}$ and melt energy was highest in July, except for Drangajökull which had similar $SW_{net}$ values in June and July. This may be associated with there being fewer impurities in the exposed bare ice at Drangajökull compared to the other main glaciers, which are closer to volcanic activity and dust hotspots.

The highest variability of $SW_{net}$ occurred in June and July, associated with the extent of bare-ice areas, driven by melt intensity in the following spring, and previous winter snow depth. In April, the cold content of winter snow limits the melt

energy available to produce melt water, and winter snow still covers the impurity-rich ice in the ablation areas. $LW_{net}$ was
negative for all months at all locations, with a slight decrease (less negative) for the latter half of the melt season (JAS).
Turbulent fluxes showed little variability between months for the averages presented, whereas SHF often had peaks associated
with events or prolonged periods in which warm air was transported, enhancing melt.

## 4.3    Impacts of volcanic eruptions and other LAP events

Inter-annual SEB variability for Icelandic glaciers was generally high. Figure 8 shows SEB anomalies as deviations from
the period mean. The 2004 eruption in Grímsvötn and the associated LAP deposits had some, though very limited, impact on
ablation, since it took place in the fall, prior to the buildup of the winter snowpack. In the following melt season (2005), impacts
of the tephra deposits were observed at Vatnajökull. For Vatnajökull, the increase in $SW_{net}$ was 15 % above the mean $SW_{net}$
energy over the period. The impacts were notable in the northern and southeastern outlets, being 20–27 % $SW_{net}$ above the
mean of the study period. In southwest Vatnajökull, $SW_{net}$ was very close to the period mean, with a 1 % increase, indicating
the limited impact of the 2004 LAP deposits. The discrimination between tephra deposits from the eruption and loading of
LAPs from other sources during the summer of 2005 is complex, and perhaps the extensive $SW_{net}$ in southeast Vatnajökull
was a combination of both, i.e., added LAPs from dust hotspots in the northern highlands during the melt season and LAPs
from the eruption. For Langjökull, Hofsjökull and Eyjafjallajökull, $SW_{net}$ was below average, indicating that the influence of
LAP deposits from the eruption was negligible. The northeastern outlet at Mýrdalsjökull had an increase in $SW_{net}$, more likely
due to dust from surrounding hotspots rather than residual effects from the eruption in 2004.

The figure shows that for the period from 2000 to 2021, with the exception of Drangajökull, the highest MJJA melt energy
was observed in 2010, associated with a warm, cloud-free summer and $SW_{net}$ amplification due to LAP depositions from
the Eyjafjallajökull eruption, generally lowering albedo. For Vatnajökull, the increase in $SW_{net}$ was 25 % above the mean
$SW_{net}$ over the study period. For southwestern Vatnajökull, the $SW_{net}$ was 33 % above the period mean, while it was 20, 29
and 16 % for the northeastern, northwestern and southeast outlets of Vatnajökull, respectively. At Hofsjökull, the increase in
$SW_{net}$ was about 35 % for the whole glacier, with the highest anomaly being 44 % for the southeastern outlet. Lower $SW_{net}$
enhancements of 29 % were observed at the northern parts of Hofsjökull, and the southwestern outlet had an increase of 36
%. The impacts for Langjökull were similar to those for Hofsjökull, with the increase in radiative forcing being higher for the
southern northeastern outlets (42 % and 43 %, respectively) but lower for the north-facing outlet (29 %). The spatial variations
in radiative forcings are mostly explained by the distribution of the volcanic ash plumes transported from Eyjafjallajökull in
mid-May 2010 (Gunnarsson et al., 2021). For Mýrdalsjökull and Eyjafjallajökull, the impacts on $SW_{net}$ had generally less
spatial variability, explained by the proximity to the LAP origin and the relative size of these glaciers. For Mýrdalsjökull, the
average short-wave radiative forcing increase was 45 %, and it was 55 % for Eyjafjallajökull. On extensive areas of these
glaciers, the tephra layer was thick enough to isolate the surface (larger than 2 cm) and limit the use of melt energy to produce
melt water.

In 2011, LAP from the May sub-glacial eruption in Grímsvötn enhanced short-wave radiative forcing, mostly influencing
the southwestern and southeastern outlets of Vatnajökull. The melt energy anomaly (compared to the average melt season)

at southwest Vatnajökull was 47 %. At the northeastern outlet, $SW_{net}$ was slightly below average (99 % of mean), but the southeast and northwest observed some melt enhancement, 10 and 7 %, respectively. For Hofsjökull and Langjökull, similar $SW_{net}$ increases were observed, ranging between 14 and 22 % and with less spatial variability than for the previous year. For Mýrdalsjökull and Eyjafjallajökull, smaller average melt enhancements were seen, 16 and 25 %, respectively. A major climatological difference between 2010 and 2011 relates to the much higher average cloud cover in 2011 influencing $SW_{in}$ and generally lower air temperatures, reducing the melt enhancement potential from LAPs in 2011 compared to 2010. For both 2010 and 2011, limited impacts on $SW_{net}$ forcing were observed for Drangajökull, indicating limited impacts of LAPs from the 2010 and 2011 eruptions.

In late April 2019, rapid melt-out of seasonal snow in the highlands was observed. This was followed by favorable conditions for airborne LAPs, from dust hotspots and pro-glacial areas, enabling vast LAP deposits on glacier surfaces, with an associated decrease in albedo and potential for enhancing radiative forcing. For Vatnajökull, $SW_{net}$ was 12 % above average, with 3, 8 and 7 % $SW_{net}$ above mean for the northeast, northwest and southeast, respectively, but 18 % for the southwest outlet. For Hofsjökull, $SW_{net}$ was 16 % above average, with 10, 20 and 14 % $SW_{net}$ above the mean for the northern, southeastern and northwestern outlets, respectively. At Langjökull the values were somewhat higher: $SW_{net}$ was 23 % above average, with 21, 20 and 25 % $SW_{net}$ above the mean for the northeastern, northwestern and southern outlets, respectively. $SW_{net}$ was 12 % above average for Mýrdalsjökull, 26 % for Eyjafjallajökull and 10 % for Drangajökull. In 2019, cloud cover was generally slightly above average (more clouds), with a colder than average spring, but a dry, warm spell in midsummer.

The onset of the 2010 and 2011 eruptions in early spring maximized their impact on melt, as the LAPs could enhance radiative forcing for almost the whole melt season while the tephra deposits in fall 2004 were quickly buried in the winter snow.

## 4.4 Melt enhancement due to volcanic eruptions and other LAP events

The impacts of the high LAP deposits in 2004, 2010, 2011 and 2019 were assessed to better understand impacts on melt energy. The effect on $SW_{net}$ forcing was estimated by comparing the SEB, assuming mean albedo for the study period (2000–2021 mean excluding 2010, 2011 and 2019 in the mean), to the energy balance estimated using the observed albedo in 2010, 2011 and 2019, utilizing the same climatology forcings for both albedo scenarios. The estimated difference in $SW_{net}$ forcing, from the observed albedo scenario and the study period mean albedo scenario, was denoted as the SW radiative forcing from LAPs ($SW_{LAP}^{f}$) and refers to the increased forcing in W m$^{-2}$ above the study period mean values. The increase in melt potential, due to the additional radiative forcing from LAPs, was defined as the direct conversion of $SW_{LAP}^{f}$ into water equivalent using latent heat of fusion ( 0.26 mm day$^{-1}$ per Wm$^{-2}$ ) and was referred to as $SW_{LAP}^{mm}$. This approach does not fully consider all physical processes: e.g., as it did not take into account the effect on albedo of different snow metamorphosis processes between years, or the timing of melt-out of impurity-rich ice; but in this comparison these processes were secondary to the overwhelming impact LAPs had on the albedo, especially in 2010 and 2011. Additionally the impacts on turbulent fluxes were ignored as they are considered negligible.

Figure 9 shows on monthly timescales how the $SW_{LAP}^f$ (converted to mm) evolved between April and September for these selected years. In 2005 the $SW_{LAP}^f$ was 4.3 W m$^{-2}$, here associated with the November 2004 eruption in Grímsvötn, yielding a 211 mm $SW_{LAP}^{mm}$ for the AMJJAS period. The sharp increase in $SW_{LAP}^{mm}$ in July was associated with tephra layers melting out of the winter snow in the lower accumulation areas. Other glaciers did not experience a $SW_{LAP}^f$ increase, with the exception of Drangajökull where it is unlikely to have been caused by LAPs from the 2004 Grímsvötn eruption. Figure 8 shows the distribution of melt energy, indicating that the southwestern outlets of Vatnajökull experienced limited impact. In a similar manner the melt potential increase for Mýrdalsjökull, mainly focused on the northeast outlet, is unlikely to have been linked to LAPs from Vatnajökull.

For Vatnajökull the $SW_{LAP}^f$ was 18.4 W m$^{-2}$ in 2010, i.e., the estimated additional radiative forcing due to LAPs compared to the long-term average for AMJJAS. This corresponds to 892 mm of $SW_{LAP}^{mm}$ for the AMJJAS period. Extensive $SW_{LAP}^f$ increase was seen for all the major glaciers in 2010 with the exception of Drangajökull. $SW_{LAP}^f$ from LAPs was 27.3, 27.7, 44.6 and 53.2 W m$^{-2}$ for Hofsjökull, Langjökull, Mýrdalsjökull and Eyjafjallajökull, respectively. As expected, the impacts were most extensive at Eyjafjallajökull and Mýrdalsjökull due to the proximity of the eruption and LAP source. Drangajökull was the exception to these extremes, with only a slight increase in $SW_{LAP}^f$, 2.5 W m$^{-2}$, meaning the impact of LAP deposits associated with the 2010 eruption was limited. Further increasing the potential impact of LAPs on melt energy, 2010 had cloud cover well below average.

The Grímsvötn eruption in 2011 had most impact on Vatnajökull, especially its southwestern outlets. The impact was similar to that in 2010, with $SW_{LAP}^f$ of 19.1 W m$^{-2}$ (925 mm of $SW_{LAP}^{mm}$). For other glaciers the impact was much less than in 2010. For Eyjafjalljökull and Mýrdalsjökull the impact was more likely related to the huge quantities of tephra deposits from the 2010 eruption than additional LAPs from the 2011 Grímsvötn eruption. For Langjökull and Hofsjökull the $SW_{LAP}^f$ was 8.6 and 8.8 W m$^{-2}$ (415 and 425 mm of $SW_{LAP}^{mm}$), respectively. As previously mentioned, the melt season in 2011 had a different climatology than the previous year, with above-average cloud cover and lower air temperatures, not fully utilizing the melt enhancement potential from the LAPs deposited. The vast quantities of tephra transported in 2010 to glaciated surfaces, as well as those deposited outside glacier-covered areas, likely had a residual effect in 2011. $SW_{LAP}^f$ was negative for Drangajökull in 2011.

The large observed LAPs in 2019 yielded a significant $SW_{LAP}^f$ for all glaciers. The $SW_{LAP}^f$ was 7.0 Wm$^{-2}$ (341 mm of $SW_{LAP}^{mm}$) at Vatnajökull, 12.9 Wm$^{-2}$ (624 mm $SW_{LAP}^{mm}$) at Langjökull, and 8.4 Wm$^{-2}$ (407 mm of $SW_{LAP}^{mm}$) at Hofsjökull. At these glaciers the $SW_{LAP}^f$ was higher in the early melt season, with less impact in July and August, partly due to frequent snowfall events in mid- and late August increasing albedo and reducing $SW_{net}$. At Drangajökull the $SW_{LAP}^f$ was the highest for the years studied, resulting in a $SW_{LAP}^f$ of 5.6 Wm$^{-2}$, yielding 245 mm of $SW_{LAP}^{mm}$.

## 5 Summary and discussion

As expected, the results show short-wave radiation as the major melt energy component and reveal variability in melt energy between glaciers and years. Water resources in Iceland rely heavily on glacier melt and are increasingly aware of the potential impacts of radiative forcing by LAP in snow and ice relative to glacier mass balance, and the impact on regional hydrology.

This study improves knowledge of the spatio-temporal variations of the surface energy balance of glaciers in Iceland and shows the importance and the possibility of incorporating remotely sensed albedo during the melt season. Albedo is often greatly impacted by external processes, not generally represented in glacier modeling, but can be observed in remotely sensed data. Although less albedo variability is expected during winter, a challenge remains due to polar darkness, as limited albedo data is available during winter to model the surface energy balance for the full hydrological year.

The impacts of tephra deposits from volcanic eruptions and dust and sand transport from pro-glacial areas (dust hot spots) were clearly seen through the radiative forcing by LAPs in snow and glacier surfaces. With future projections of less seasonal snow during winter due to climate warming, earlier melt-out of seasonal snow in spring exposing pro-glacier areas (dust hot spots) will likely become more frequent enhancing melt (Björnsson et al., 2018). Here, estimates show that melt enhancement due to high LAPs was 10—50% higher than average, depending on glacier investigated, greatly impacting glacier summer mass balance. Estimations of future glacier development, need to incorporated these processes for improved estimation of the SEB.

Since climate in Iceland is driven by oceanic and atmospheric circulations, annual climate variability is high, reflected in complex processes in large-scale global climate dynamics. For the study period, 2000–2021, all input and output data were checked for monotonic trends using the non-parametric Mann–Kendall test in terms of the total change of a least-square fit. The probability of rejecting the null hypothesis when the null hypothesis is true was estimated at $\alpha$ as 0.05. In most cases the high climate variability, on both monthly and annual time scales yields non significant trends in the data over the study period. Trends over longer timescales for glacier runoff and increased mass loss of Icelandic glaciers are obvious and have been confirmed in other studies (Björnsson et al., 2018; Schmidt et al., 2020; Aðalgeirsdóttir et al., 2020; Noël et al., 2022). Gunnarsson et al. (2021) found significant positive trends for surface albedo for the 2000–2019 period in parts of NE and NW Vatnajökull. Very low albedo values in NE and NW Vatnajökull in summer 2021 have mostly eradicated the trend, if the period is extended to 2021. Since the data only spans 20 years, statistical interpretation such as trends should be treated with care.

Figures B6 to B11 show annual anomalies variations (MJJA) for selected climate variables over the study period 2000–2021. $SW_{net}$ is highly modulated by cloud cover and surface albedo and the variability was reflected in cloud cover and surface albedo variability. High cloud cover in 2015 and 2018 coincides with the only years with positive observed surface mass balance during the study period for Vatnajökull (Pálsson et al., 2020). As an example of the complex relations between Iceland climate and oceanic and atmospheric circulations, Keil et al. (2020) have suggested that more low-level clouds are being produced due to cooler sea surface temperatures at the North Atlantic Warming Hole (56° to 50°N and 33° to 39°W), leading to reductions in incoming solar radiation and further surface cooling.

## 5.1 Uncertainty sources and limitations

Remotely sensed MODIS albedo data availability is impacted by persistent cloud cover. Since gap-filled albedo was based on temporal aggregation potentially short-lived snowfall events during summer could be missed within extended periods of cloud cover, increasing albedo. This would results in overestimation of melt during the period. On a melt season timescale these effects were likely limited although more prominent on shorter timescales. Tephra deposits from volcanic eruptions can, in proximity to the eruption source, have enough thickness to isolate the surface from melt energy. The results presented here

assume that areas with low albedo values post-eruptions in proximity to eruption sources do not have isolating capabilities which introduces some uncertainty. This is partly due to the remotely sensed albedo workflow, as it can not identity and discriminate between areas that isolate the surface and those that do not. Future work, e.g., spectral unmixing might contribute to spatial estimations of isolation capabilities of the tephra layers. Further discussion of the albedo data processing limitations are found in Gunnarsson et al. (2021).

Currently, no spatio-temporal data set exists with areal estimates of tephra layer thickness that could have been incorporated into the workflow. In these areas, where the surface becomes isolated, the surface temperature could rise above the 273.15 K constrain of a melting surface covered with ice or snow, impacting the estimation of outgoing long-wave radiation, and turbulent heat fluxes. This also influences one of the main assumptions for the SEB model, that closes the SEB by iteratively solving for surface temperature, indicate availability of melt energy with solutions higher than 273.15 K. Since limited data were available

to fully estimate where isolation might have occurred, and more complex modeling is needed to fully represent the problem, all energy balance components were considered to contribute to melt energy for all areas and times. Since outgoing long-wave radiation was calculated based on the Stefan–Boltzmann law assuming broadband emissivity of snow and ice, isolating tephra layers with would influence these assumptions and the resulting LW↑ estimate with changes in surface emissivity.

Volcanic tephra deposits were thick enough to isolate parts of the surface at Eyjafjallajökull and Mýrdalsjökull in 2010 and

535 very probably in the following year or years, causing partial isolation of the surface. In 2011, parts of the glacier surface around Grímsvötn in Vatnajökull were isolated, but as this occurred mostly in the accumulation area the post-years effect was very likely limited.

Various limitations to turbulent fluxes estimations using the bulk aerodynamic approach exist as many assumptions and simplifications were utilized to estimate the complex interactions between the atmosphere and the glacier surface. Thick enough

tephra deposits could impact the surface roughness values ($z_0$) that impact the turbulent flux estimation. Our experience suggests that the deposition of airborne LAP (non-volcanic origin) to glacier surfaces do not accumulate to the extent that the surface properties shifts from snow/ice to soil, allowing for surface temperature above 273.15 K. Various studies have estimated the sensitivity of $z_0$ and impacts on the estimated turbulent fluxes; underestimated $z_0$ values will result in underestimation of turbulent fluxes and vice versa (Denby and Greuell, 2000; Schmidt et al., 2017). In this study, $z_0$ was estimated for each pixel

with a general usage of single values for ice and snow based on surface albedo values. Surface classification, discriminating between ice and snow, assuming surface albedo higher than 0.45 as snow and lower than 0.45 as ice has limitations as well.

In reality these values likely had more spatial and temporal variability depending on glaciers and glacier outlets in Iceland but limited studies exist. This is a research topic for future studies.

The model meteorological forcing data was combined from three different, but similar, WRF configurations. Although the bulk of the period was from the WRF RAV2 configuration, externally forced with ECMWF ERA-Interim data, the more recent data, ICEB and FCST, was forced with the National Centers for Environmental Prediction Global Forecast System (GFS). This yields that the combined forcing datasets might not be fully consistent. Although the validation results show similar behavior of the variables validated, both for the whole period and individual months during the melt season, there was much less data available for ICEB and FCST for thorough validation. Alternative external forcing could also result in different dynamical behaviors, such as storm tracks, moisture transport and other complex climate dynamics represented differently for different forcing data, impacting melt. Quantification of the effects of combining forcing dataset is challenging with no extensive overlapping periods.

In the SEB model the sub-surface heat flux (cold content of the snow) was assumed to be zero as was energy from precipitation. This is a source of uncertainty. Since the snowpack accumulation during winter was not modeled the cold content was not easily estimated although Icelandic glaciers are temperate glaciers. In the study the energy contribution from cold content was assumed to be zero, i.e. no contribution to the energy balance. Observed temperature in spring surface mass balance data indicate that cold content was not a major energy source/sink, although it provides regulation of the melt energy during the warming and ripening phase of the snow pack (Jennings et al., 2018; Helgason, 2020)(unpublished data). The assumption of zero sub-surface heat flux for AWS data has been applied in many recent studies of energy balance and surface melt for Icelandic glaciers (e.g., Guðmundsson et al., 2005, 2009; Wittmann et al., 2017; Schmidt et al., 2017).

The validation between the observations and met forcings incorporate uncertainties that impact the calculated energy balance. Since short-wave radiation forced at the surface $\text{SW}_{net}$ were the major drivers of melt energy, both uncertainties in the $\text{SW}_{in}$ and estimated surface albedo contribute to over- or underestimation of melt energy. Biases in air temperature were generally negative, suggesting a model overestimation of air temperature, resulting in overestimation of turbulent fluxes, and thus with melt energy overestimation.

Observed data in complex and often harsh weather conditions can also include errors from instrumentation. When validating modeling results and remotely sensed products with observations, it is important to consider these sources of error, both in time and space. When validating a larger spatial footprint data, such as 500 m down-scaled forcing and MODIS albedo, and comparing them to point observations from met stations this should be considered as high spatial variability and sub-pixel variability often exists for Icelandic glaciers, especially in the bare-ice areas.

## 6  Conclusions

In this study, melt-season SEB for Icelandic glaciers was estimated using high-resolution meteorological climate forcing and remotely sensed glacier surface albedo from the MODIS sensor for melt seasons 2000–2021 at 500 m spatial resolution. The

calculation framework was applied to all glaciers in Iceland larger than 8 km$^2$, but results are presented for the largest glaciers,
Vatnajökull, Langjökull, Hofsjökull, Mýrdalsjökull, Eyjafjallajökull and Drangajökull.

The main results show large seasonal and inter-annual variability in SEB for Icelandic glaciers. The variability was influenced by high climate variability, LAPs from tephra deposits from volcanic eruptions and dust hotspots from sources and pro-glacial areas close to the glaciers. Dust hotspots are subject to wind erosion and production of LAPs that can be transported over long distances.

The high variability meant that no significant trends were found, either in data driving the model or in the model output data. Details of spatio-temporal patterns were obtained, as well as relations to elevation and distribution of melt energy with elevation between years. The main energy melt source was from short-wave radiation modulated by surface albedo and cloud cover, which is in good agreement with previous studies.

The impacts of volcanic eruptions during the period (in 2004, 2010 and 2011) through the effect of dust and tephra deposits
on radiative forcing were estimated by modeling the short-wave radiative forcing under observed albedo scenarios during the relevant periods and comparing them to alternative evolution of albedo. The impacts were assessed by estimating the additional energy forced for surface melting, with up to 55 % additional melt energy forcing being found compared to the study period average. Radiative forcing due to LAPs in 2019 deriving from extensive airborne dust and sand deposits was also estimated, yielding a significant impact on the energy balance, with melt energy increasing significantly compared to the study-period
average albedo development under the same 2019 climate.

The methodology applied in the study, based on MODIS products and climate forcing data, can be utilized in near-real time to assess the impacts of LAPs associated with volcanic eruption and dust storm deposits in ice and snow surfaces, providing insight into melt enhancements. It also allows for short-term as well as long-term monitoring of SEB variations for glaciers in Iceland.

*Data availability.*  MODIS data are available from https://nsidc.org/data (Hall and Riggs, 2016a, b). Geospatial data for Iceland are available from the National Land Survey of Iceland at https://atlas.lmi.is. Glacier automatic weather station data, climate forcings and SEB output are available upon request.

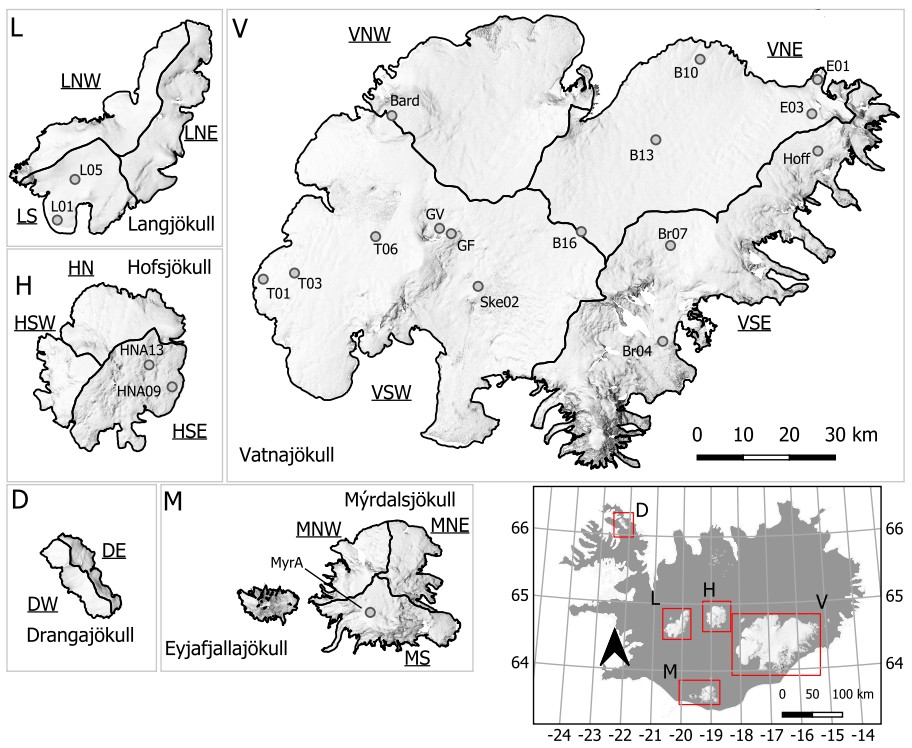

**Figure 1.** Location map of Icelandic glaciers in this study. Vatnajökull, Langjökull, Hofsjökull, Mýrdalsjökull, and Drangajökull are divided into main ice flow basins for further detailed analysis. These delineated areas are annotated with underlined text (e.g., NW for northwest). Locations of automatic weather stations (AWSs) are shown with grey dots. Details of the AWSs are given in Table A1. Topographical properties of the ice caps and their sub-areas are listed in Table 1. The background is shaded relief of Lidar-surveyed glacier DEMs (Jóhannesson et al., 2013) and the catchment delineations are from Magnússon et al. (2016b), for Drangajökull, Björnsson (1988) for Hofsjökull, Björnsson et al. (2000a) for Mýrdalsjökull, and Pálsson et al. (2015, 2020) for Langjökull and Vatnajökull, respectively. The scale for Vatnajökull (V) applies for all glacier maps (L, H, D, M and V)

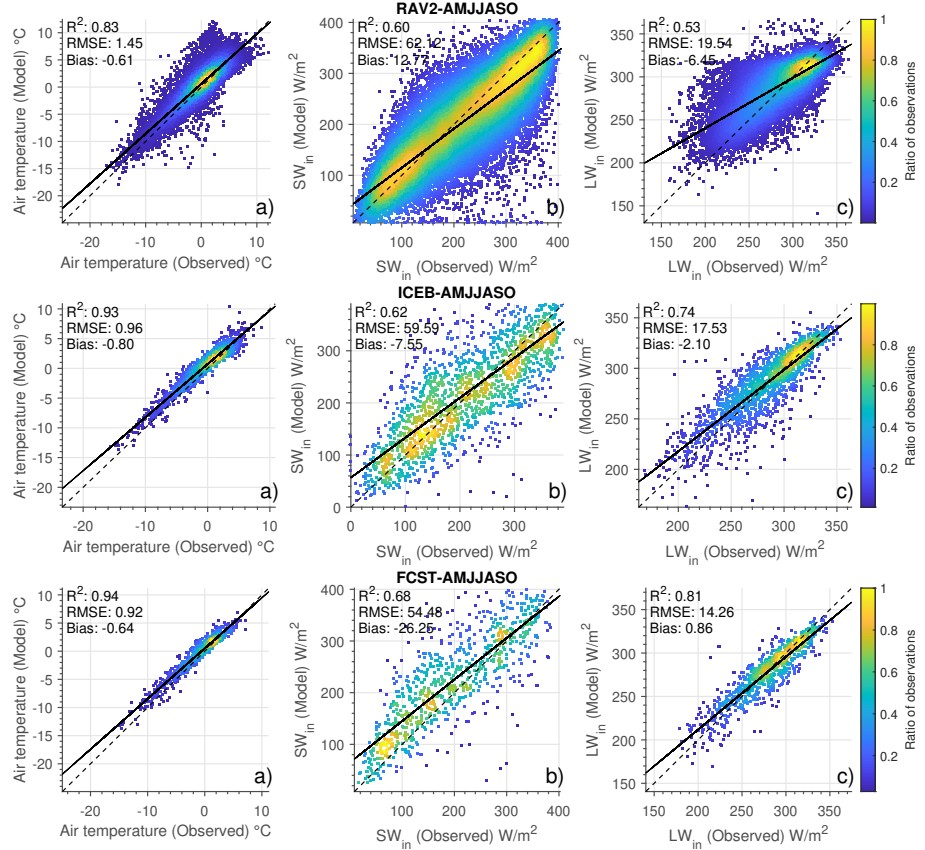

**Figure 2.** Comparison of the downscaled daily model forcings, 2 m air temperature (a), incoming solar radiation (SW↓) (b) and incoming long-wave radiation (LW↓) (c), with ground observations from the GAWS network. Data is shown for averages from April through October (AMJJASO). Color shows the normalized (0–1) density distribution of data. Different rows indicate different data sources (RAV2, ICEB and FCST) used in the comparison, see Section 3.3. Dotted black line shows 1:1 and black line the calculated linear fit to the data. Further details are in Table 2.

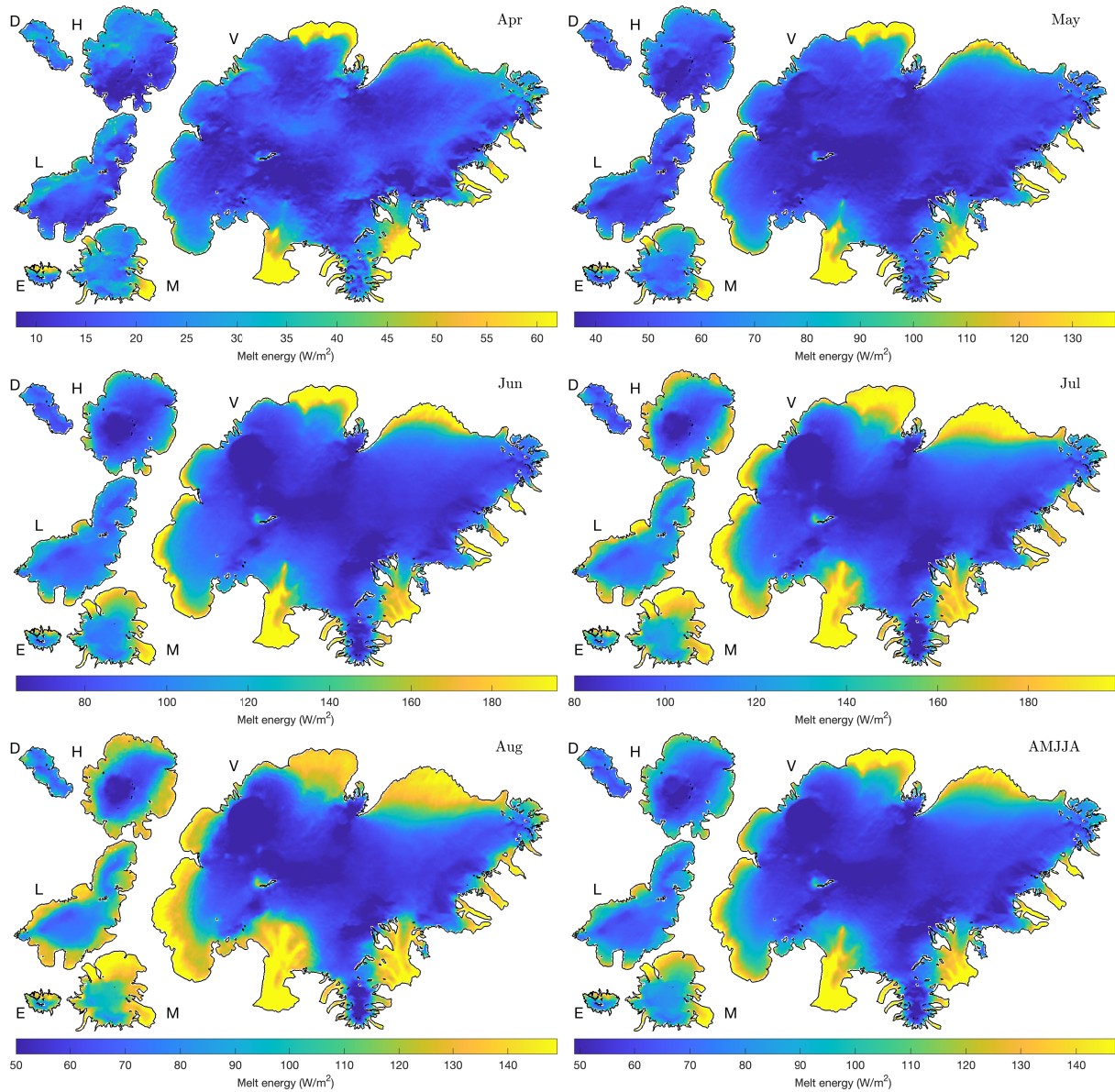

**Figure 3.** Spatial patterns of mean melt energy for the period 2000–2019 (AMJJA). D: Drangajökull; H: Hofsjökull; V: Vatnajökull; L: Langjökull; E: Eyjafjallajökull; M: Mýrdalsjökull. Note that the color scale varies between months.

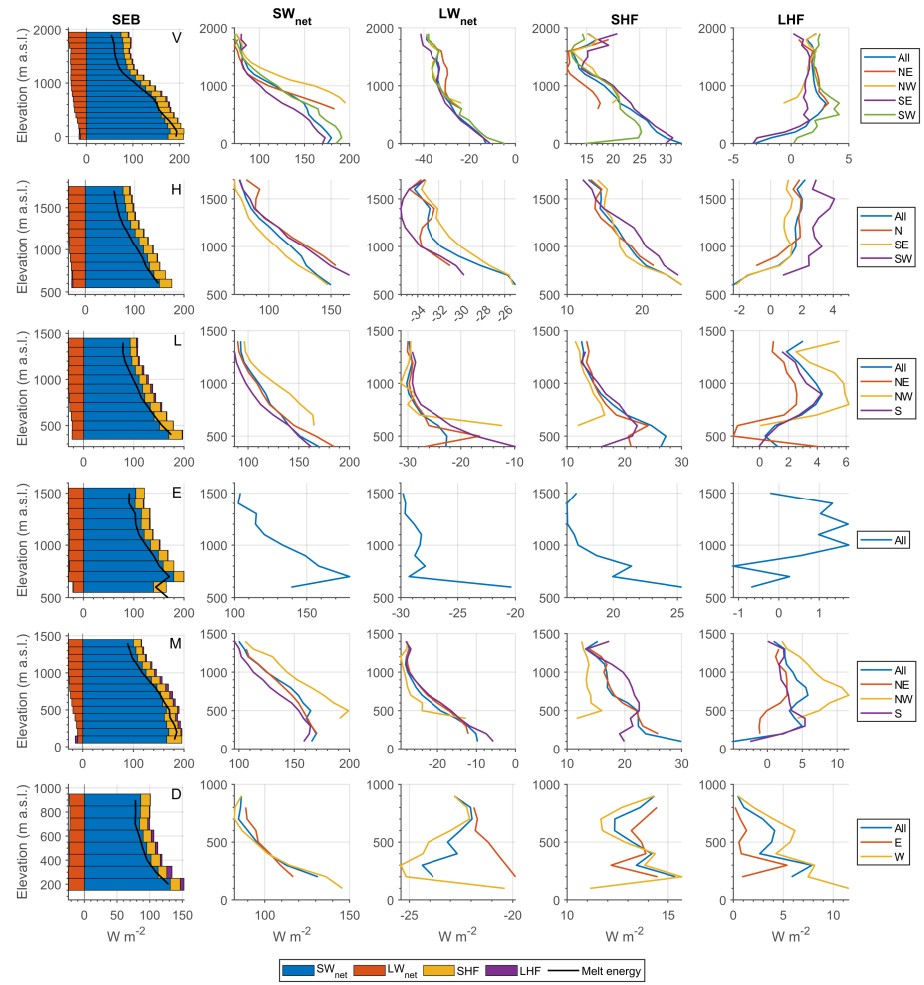

**Figure 4.** Variation of surface energy balance components with elevation (100 m elevation bins). The first column of images shows the average MJJA energy balance by elevation for the whole glaciers. The other columns show individual SEB components for the glaciers and their main sub-areas as a function of elevation. (Sub-areas are defined in Fig. 1). V: Vatnajökull; H: Hofsjökull; L: Langjökull; E: Eyjafjallajökull; M: Mýrdalsjökull; D: Drangajökull. Note that the horizontal scale varies between panels. $SW_{net}$ is the incoming short-wave radiation, $LW_{net}$ is the net long-wave radiation, SHF is the sensible heat flux and LHF the latent heat flux.

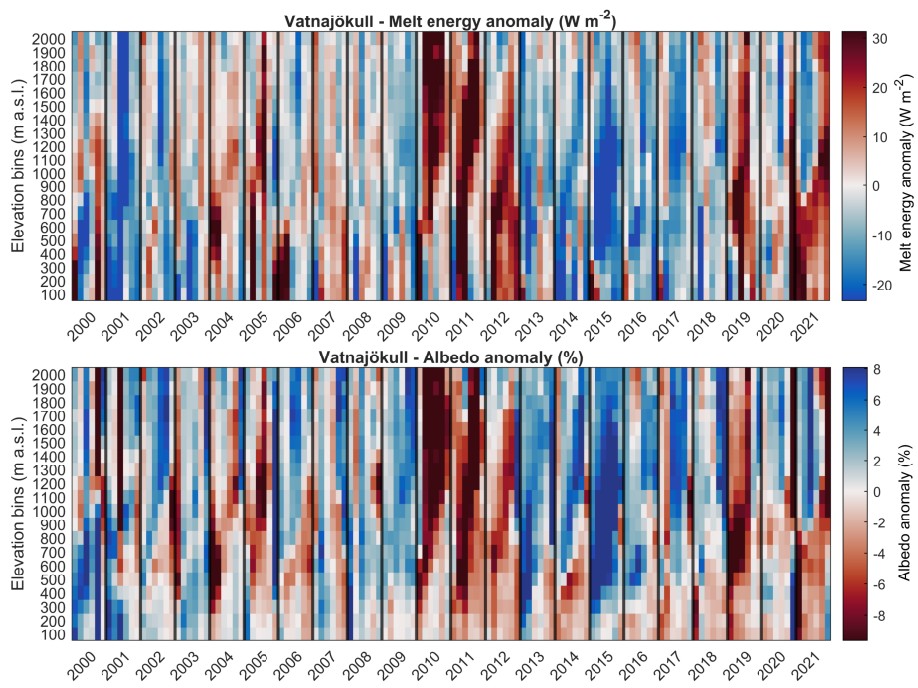

**Figure 5.** Variation of monthly average melt energy (W m$^{-2}$, upper panel) and monthly albedo anomalies (lower panel) for Vatnajökull. Elevation (vertical axis) is in bins of 100 m and the horizontal axis shows monthly data for each year from April to September. Black vertical lines separate the years. Figures B1 to B5 in the Appendix are similar figures for other glaciers.

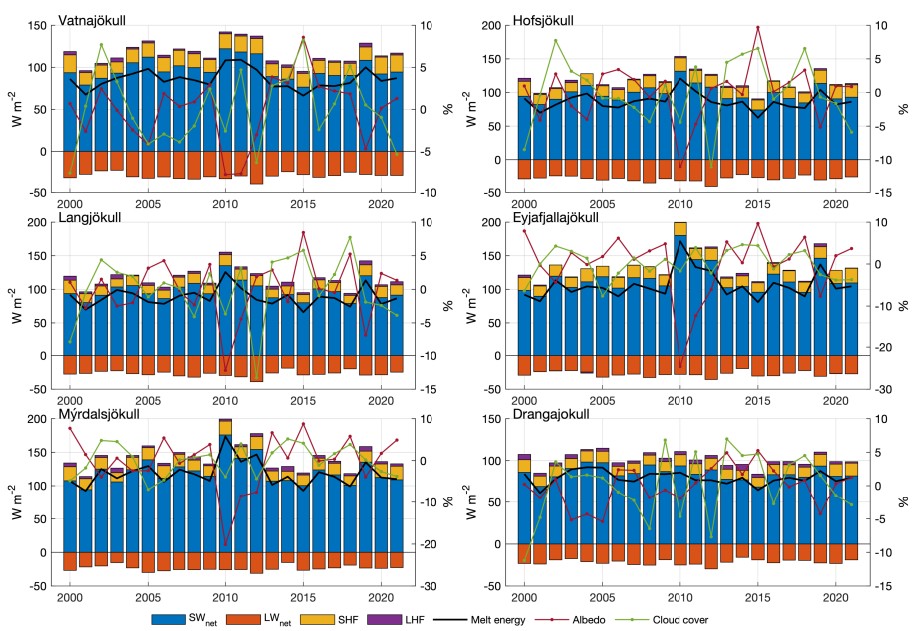

**Figure 6.** Surface energy balance sources (colored bars) and the available melt energy (solid black line) for the period MJJA 2000–2021 (left vertical axis). The melt season mean albedo (purple) and cloud cover (green) for each glacier is shown as deviations from the period mean (right vertical axis). $SW_{net}$ is the incoming short-wave radiation, $LW_{net}$ is the net long-wave radiation, SHF is the sensible heat flux and LHF the latent heat flux

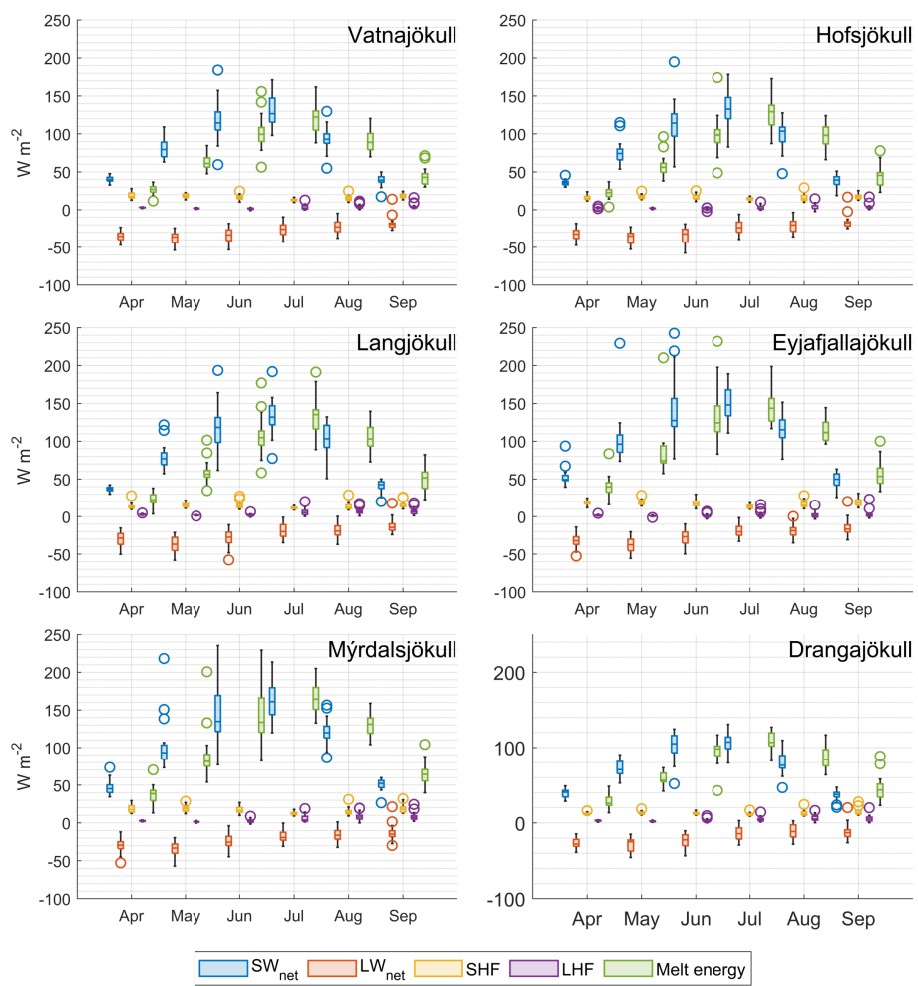

**Figure 7.** The monthly average distribution of surface net energy balance components and melt energy for the glaciers studied. $SW_{net}$ is the incoming short-wave radiation, $LW_{net}$ is the net long-wave radiation, SHF is the sensible heat flux, LHF the latent heat flux. The line inside of each box is the sample median, the top and bottom edges of each box are the upper and lower quantiles (0.25 and 0.75), respectively. The whiskers that extend above and below each box connects the upper/lower quantiles to the nonoutlier maximum/minimum. Circles represent outliers.

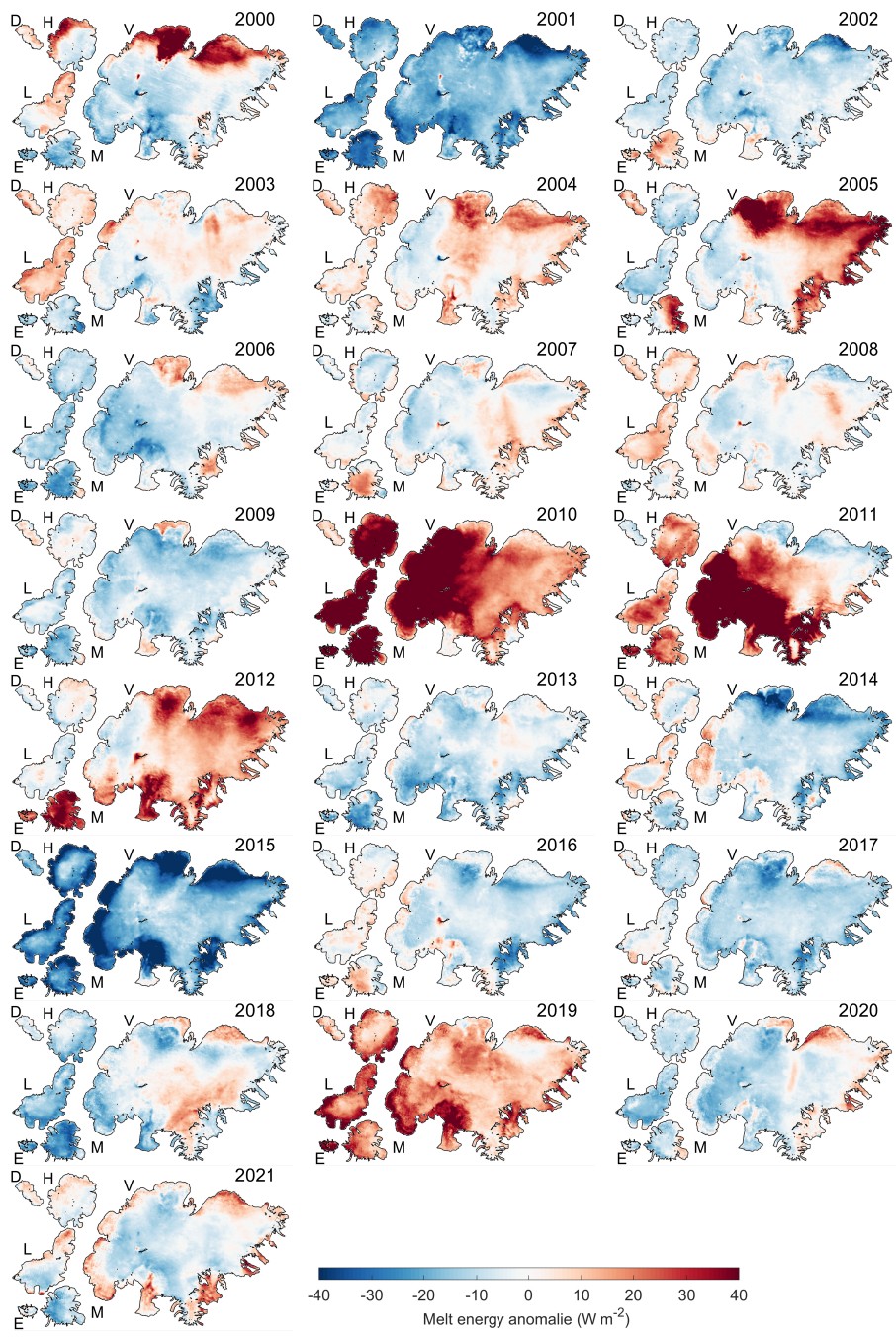

**Figure 8.** Surface energy balance anomalies from the mean for MJJA 2000–2021. Red colors indicate average melt energy above average (more potential melt energy) while blue colors denote surface energy balance below average.

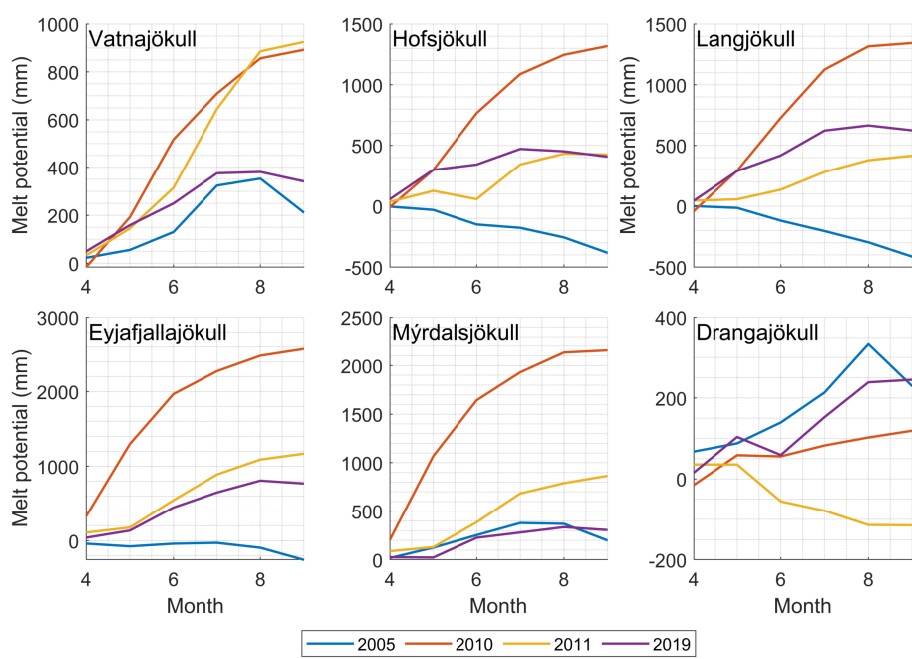

**Figure 9.** Estimated increase in melt potential (mm of water) due to the effect LAPs had on the surface energy balance in 2005, 2010, 2011 and 2019. Data are shown as the increase in cumulative monthly melt potential due to LAPs, i.e., the difference in melt using historical average albedo (2000–2021 mean excluding 2010, 2011 and 2019 in the mean) and observed albedo for the selected years using the same climatological forcings. Note that the vertical scale varies between panels.

**Table 1.** Topographic properties of the 6 main glacier catchments and their 15 sub-areas. Id column refers to the sub-glacier areas shown in Figure 1. Ratio defines the area percentage of each sub-area with respect to the relevant glacier total area. Elevation data are from Jóhannesson et al. (2013) and glacier area from Hannesdóttir et al. (2020).

| Id | Glacier | Area | $z_{mean}$ | $z_{max}$ | $z_{min}$ | Ratio |
|---|---|---|---|---|---|---|
| | | km$^2$ | m a.s.l. | m a.s.l. | m a.s.l. | - |
| | Vatnajökull | 7881 | 1223 | 2030 | 0 | - |
| VNE | NE | 1669 | 1229 | 1888 | 629 | 21 % |
| VNW | NW | 1239 | 1406 | 1988 | 729 | 15 % |
| VSE | SE | 1952 | 1066 | 2030 | 0 | 25 % |
| VSW | SW | 3051 | 1225 | 1994 | 61 | 39 % |
| | Hofsjökull | 852 | 1252 | 1789 | 624 | - |
| HN | N | 287 | 1289 | 1789 | 830 | 34 % |
| HSE | SE | 402 | 1200 | 1789 | 637 | 47 % |
| HSW | SW | 162 | 1346 | 1789 | 735 | 19 % |
| | Langjökull | 896 | 1102 | 1435 | 419 | - |
| LNE | NE | 304 | 1090 | 1435 | 419 | 34 % |
| LNW | NW | 307 | 1137 | 1435 | 620 | 35 % |
| LS | S | 284 | 1020 | 1400 | 444 | 31 % |
| | Mýrdalsjökull | 562 | 1000 | 1485 | 118 | - |
| MNE | NE | 151 | 893 | 1377 | 223 | 27 % |
| MNW | NW | 149 | 1051 | 1455 | 414 | 26 % |
| MS | S | 271 | 997 | 1485 | 118 | 47 % |
| | Drangajökull | 144 | 658 | 914 | 213 | - |
| DE | E | 52 | 653 | 872 | 297 | 37 % |
| DW | W | 92 | 655 | 914 | 186 | 62 % |
| Eyj | Eyjafjallajökull | 79 | 1156 | 1564 | 294 | - |

**Table 2.** Summary statistics for daily incoming solar radiation (SW↓), incoming long-wave radiation (LW↓), and air temperature from different WRF configurations validated with ground observations. No. sites refers to the number of stations that were available for comparison purposes for each period. All results, for all months and the three variables SW↓, LW↓ and air temperature had a significant relationship ($p < 0.05$).

| RAV2 data | T (°C) | | | SW↓ (W m$^{-2}$) | | | LW↓(W m$^{-2}$) | | | Period |
|---|---|---|---|---|---|---|---|---|---|---|
| No. sites | RMSE | $R^2$ | Bias | RMSE | $R^2$ | Bias | RMSE | $R^2$ | Bias | Month |
| 20 | 1.09 | 0.83 | -0.65 | 55.61 | 0.63 | 14.36 | 13.44 | 0.63 | -6.50 | AMJJASO |
| 4 | 1.73 | 0.84 | -1.07 | 82.21 | 0.26 | 14.98 | 9.68 | 0.44 | -16.33 | Apr |
| 18 | 1.10 | 0.88 | -0.86 | 55.13 | 0.42 | 18.58 | 13.83 | 0.61 | -13.31 | May |
| 20 | 0.90 | 0.75 | -0.66 | 61.32 | 0.45 | 23.42 | 12.79 | 0.63 | -6.74 | Jun |
| 20 | 0.99 | 0.56 | -0.50 | 57.92 | 0.49 | 25.50 | 12.57 | 0.61 | -2.86 | Jul |
| 19 | 1.06 | 0.59 | -0.60 | 47.19 | 0.55 | 9.76 | 12.48 | 0.60 | -3.20 | Aug |
| 17 | 1.11 | 0.82 | -0.53 | 34.42 | 0.50 | -4.91 | 13.33 | 0.55 | -4.60 | Sep |
| 12 | 0.77 | 0.94 | -0.43 | 27.81 | 0.41 | -6.52 | 11.31 | 0.55 | -4.99 | Oct |
| ICEB data | T (°C) | | | SW↓ (W m$^{-2}$) | | | LW↓(W m$^{-2}$) | | | Period |
| No. sites | RMSE | $R^2$ | Bias | RMSE | $R^2$ | Bias | RMSE | $R^2$ | Bias | Month |
| 8 | 0.83 | 0.94 | -0.84 | 48.28 | 0.67 | -4.78 | 14.92 | 0.78 | -3.29 | AMJJASO |
| 2 | 0.95 | 0.94 | -0.97 | 39.70 | 0.46 | 2.33 | 14.90 | 0.80 | -13.84 | Apr |
| 8 | 0.88 | 0.93 | -1.15 | 51.75 | 0.39 | -6.26 | 15.64 | 0.70 | -5.09 | May |
| 8 | 0.74 | 0.92 | -0.97 | 47.38 | 0.64 | -4.06 | 12.03 | 0.81 | -3.48 | Jun |
| 8 | 0.66 | 0.75 | -0.45 | 45.04 | 0.68 | 10.46 | 12.49 | 0.78 | -2.11 | Jul |
| 7 | 0.63 | 0.80 | -0.60 | 44.22 | 0.55 | -16.47 | 16.18 | 0.58 | 3.69 | Aug |
| 5 | 0.84 | 0.88 | -0.96 | 30.00 | 0.49 | -14.09 | 15.62 | 0.73 | -3.43 | Sep |
| FCST data | T (°C) | | | SW↓ (W m$^{-2}$) | | | LW↓(W m$^{-2}$) | | | Period |
| No. sites | RMSE | $R^2$ | Bias | RMSE | $R^2$ | Bias | RMSE | $R^2$ | Bias | Month |
| 7 | 0.90 | 0.92 | -0.85 | 47.38 | 0.62 | -16.28 | 12.26 | 0.78 | -3.95 | AMJJASO |
| 1 | 0.97 | 0.95 | -1.13 | 25.05 | 0.80 | -0.36 | 7.50 | 0.92 | -11.63 | Apr |
| 7 | 0.86 | 0.92 | -0.97 | 45.62 | 0.46 | -27.75 | 12.58 | 0.77 | -4.38 | May |
| 5 | 0.70 | 0.89 | -0.79 | 41.55 | 0.62 | -16.14 | 9.42 | 0.86 | -0.07 | Jun |
| 5 | 0.66 | 0.60 | -0.58 | 37.63 | 0.72 | -22.63 | 9.80 | 0.86 | 3.05 | Jul |
| 5 | 0.63 | 0.68 | -0.27 | 37.03 | 0.61 | -36.67 | 11.47 | 0.81 | 6.05 | Aug |
| 5 | 0.70 | 0.92 | -0.43 | 24.78 | 0.65 | -24.11 | 11.78 | 0.84 | 1.54 | Sep |

**Appendix A:  Glacier weather stations location**

**Table A1.** Summary statistics and location information of meteorological stations. Figure 1 maps the location. The three last columns show the number of daily observations available for validation purposes for each variable used.

| Latitude | Longitude | Elevation | Site name | Number of air temperature measurements | Number of incoming short-wave measurements | Number of incoming long-wave measurements |
|---|---|---|---|---|---|---|
| 64.538 | 15.597 | 1141 | Hoff | 1688 | 1774 | 0 |
| 64.514 | 20.450 | 588 | L01 | 2246 | 2254 | 2254 |
| 64.302 | 17.153 | 1207 | Ske02 | 37 | 39 | 39 |
| 64.728 | 16.111 | 779 | B10 | 3224 | 3296 | 3215 |
| 64.575 | 16.328 | 1216 | B13 | 2043 | 2725 | 2338 |
| 64.402 | 16.681 | 1526 | B16 | 2575 | 2730 | 2569 |
| 64.417 | 17.319 | 1405 | Grímsvötn | 2687 | 791 | 0 |
| 64.182 | 16.335 | 528 | Br04 | 597 | 600 | 0 |
| 64.368 | 16.282 | 1242 | Br07 | 395 | 397 | 0 |
| 64.325 | 18.117 | 771 | T01 | 483 | 567 | 567 |
| 64.336 | 17.976 | 1068 | T03 | 1943 | 2586 | 2094 |
| 64.404 | 17.608 | 1466 | T06 | 2538 | 2632 | 1691 |
| 64.639 | 17.522 | 1945 | Bard | 1509 | 898 | 0 |
| 64.406 | 17.267 | 1724 | Grímsfjall | 2495 | 1324 | 0 |
| 63.611 | 19.158 | 1345 | MyrA | 385 | 413 | 0 |
| 64.594 | 20.374 | 1095 | L05 | 2536 | 2544 | 2544 |
| 64.770 | 18.543 | 840 | HNA09 | 292 | 307 | 307 |
| 64.813 | 18.648 | 1235 | HNA13 | 294 | 307 | 307 |
| 64.677 | 15.581 | 766 | E01 | 106 | 121 | 121 |
| 64.611 | 15.615 | 1190 | E03 | 115 | 122 | 122 |

**Table A2.** Summary statistics for daily outgoing solar radiation (SW↑). outgoing long-wave radiation (LW↑). and relative humidity (RH) from different WRF configurations validated with ground observations. No. sites refers to the number of stations that were available for comparison purposes for each period.

| RAV2 data | RH (%) | | | LW↑ (W m$^{-2}$) | | | SW↑(W m$^{-2}$) | | | Period |
|---|---|---|---|---|---|---|---|---|---|---|
| No. sites | RMSE | $R^2$ | Bias | RMSE | $R^2$ | Bias | RMSE | $R^2$ | Bias | Month |
| 18 | 4.77 | 0.58 | 5.31 | 7.69 | 0.29 | 3.21 | 53.65 | 0.63 | -9.68 | AMJJASO |
| 3 | 5.20 | 0.61 | 6.16 | 8.78 | 0.58 | -3.03 | 52.85 | 0.16 | 4.62 | Apr |
| 18 | 4.96 | 0.66 | 5.56 | 8.23 | 0.49 | -0.11 | 51.12 | 0.40 | -16.57 | May |
| 18 | 4.75 | 0.56 | 5.68 | 7.57 | 0.13 | 5.43 | 57.18 | 0.55 | -10.98 | Jun |
| 18 | 4.58 | 0.49 | 5.72 | 7.41 | 0.02 | 5.92 | 54.42 | 0.56 | -3.61 | Jul |
| 17 | 4.64 | 0.51 | 4.97 | 6.90 | 0.03 | 4.16 | 50.38 | 0.49 | -7.33 | Aug |
| 15 | 4.62 | 0.60 | 4.38 | 7.47 | 0.33 | 0.40 | 49.68 | 0.34 | -12.61 | Sep |
| 9 | 4.60 | 0.55 | 4.53 | 7.80 | 0.43 | -0.04 | 49.43 | 0.26 | -12.53 | Oct |

| ICEB data | RH (%) | | | LW↑ (W m$^{-2}$) | | | SW↑(W m$^{-2}$) | | | Period |
|---|---|---|---|---|---|---|---|---|---|---|
| No. sites | RMSE | $R^2$ | Bias | RMSE | $R^2$ | Bias | RMSE | $R^2$ | Bias | Month |
| 8 | 4.77 | 0.65 | 6.18 | 6.20 | 0.76 | -0.78 | 43.72 | 0.64 | -21.00 | AMJJASO |
| 2 | 3.46 | 0.81 | 8.05 | 5.05 | 0.92 | -3.92 | 32.77 | 0.50 | 7.68 | Apr |
| 8 | 4.74 | 0.56 | 8.12 | 6.76 | 0.82 | -1.82 | 52.97 | 0.40 | -29.39 | May |
| 8 | 4.26 | 0.76 | 7.13 | 6.87 | 0.47 | 1.76 | 51.12 | 0.49 | -30.95 | Jun |
| 8 | 4.82 | 0.64 | 5.16 | 5.01 | 0.08 | -0.83 | 34.51 | 0.72 | -14.32 | Jul |
| 8 | 4.52 | 0.60 | 5.58 | 5.72 | 0.38 | 0.19 | 29.84 | 0.65 | -16.71 | Aug |
| 7 | 4.63 | 0.64 | 4.23 | 4.78 | 0.82 | -3.66 | 23.41 | 0.66 | -16.96 | Sep |
| 5 | 3.47 | 0.38 | 8.01 | 4.90 | 0.89 | -4.92 | 6.87 | 0.93 | -4.99 | Oct |

| FCST data | RH (%) | | | LW↑ (W m$^{-2}$) | | | SW↑(W m$^{-2}$) | | | Period |
|---|---|---|---|---|---|---|---|---|---|---|
| No. sites | RMSE | $R^2$ | Bias | RMSE | $R^2$ | Bias | RMSE | $R^2$ | Bias | Month |
| 7 | 4.29 | 0.78 | 9.17 | 4.01 | 0.92 | -2.41 | 35.38 | 0.66 | -21.45 | AMJJASO |
| 0 | - | - | - | - | - | - | - | - | - | Apr |
| 1 | 4.65 | 0.75 | 10.81 | 5.89 | 0.83 | 0.95 | 28.52 | 0.49 | -51.46 | May |
| 7 | 2.33 | 0.95 | 9.83 | 4.53 | 0.71 | -1.39 | 70.02 | -0.03 | -38.88 | Jun |
| 5 | 5.64 | 0.75 | 7.16 | 1.84 | 0.24 | -3.17 | 11.30 | 0.49 | -4.85 | Jul |
| 5 | 3.93 | 0.70 | 9.61 | 2.55 | 0.06 | -4.24 | 8.02 | 0.11 | 1.75 | Aug |
| 5 | 5.11 | 0.43 | 7.21 | 1.94 | 0.76 | -4.93 | 23.08 | 0.65 | -20.82 | Sep |
| 0 | - | - | - | - | - | - | - | - | - | Oct |

## Appendix B: Melt energy and albedo variability with elevation

*Author contributions.* AG conceived and designed the study, performed the analyses, and prepared the manuscript. SMG and FP contributed to the study design, interpretation of the results, and writing of the manuscript.

*Competing interests.* The authors declare that they have no conflict of interest.

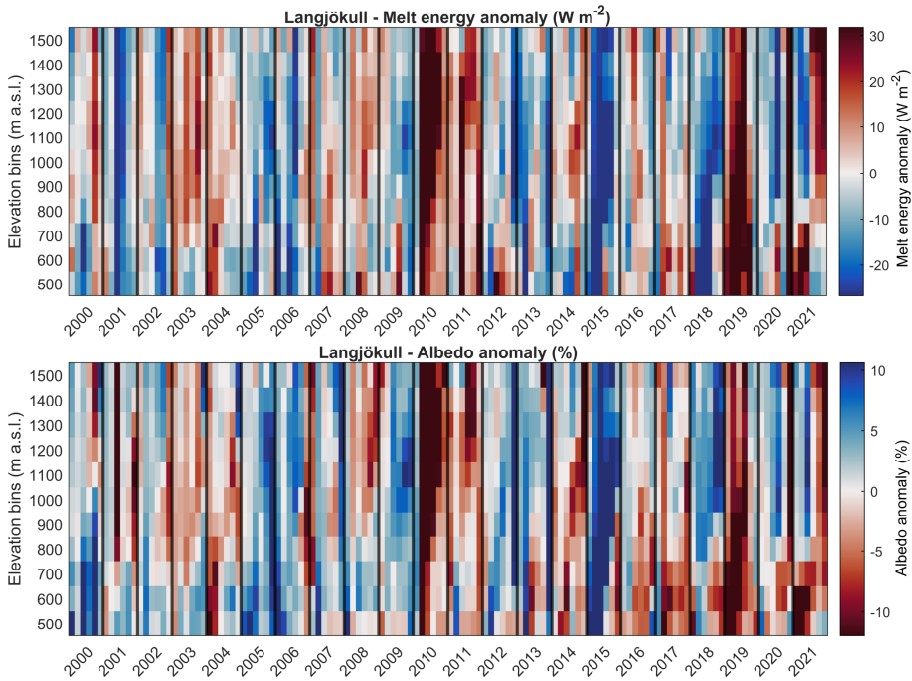

**Figure B1.** Distribution of melt energy and albedo anomalies (W m$^{-2}$) with elevation for Langjökull. Vertical axis shows elevation bins in 100 m intervals and horizontal axis shows monthly data for each year from April to September. Black vertical lines separate.

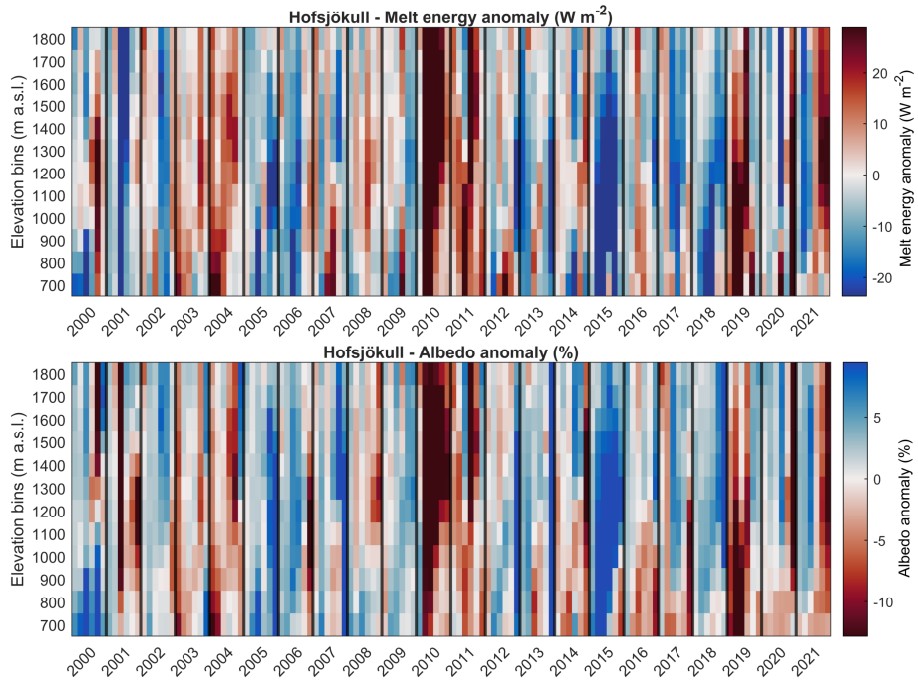

**Figure B2.** Distribution of melt energy and albedo anomalies (W m$^{-2}$) with elevation for Hofsjökull. Vertical axis shows elevation bins in 100 m intervals and horizontal axis shows monthly data for each year from April to September. Black vertical lines separate.

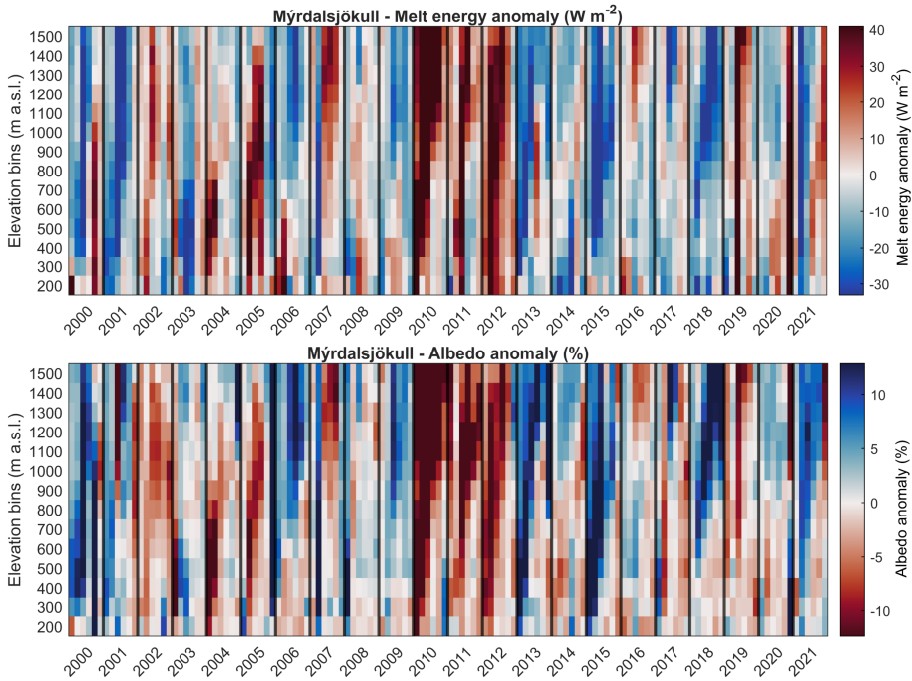

**Figure B3.** Distribution of melt energy and albedo anomalies (W m$^{-2}$) with elevation for Mýrdalsjökull. Vertical axis show elevation bins in 100 m intervals and horizontal axis shows monthly data for each year from April to September. Black vertical lines separate.

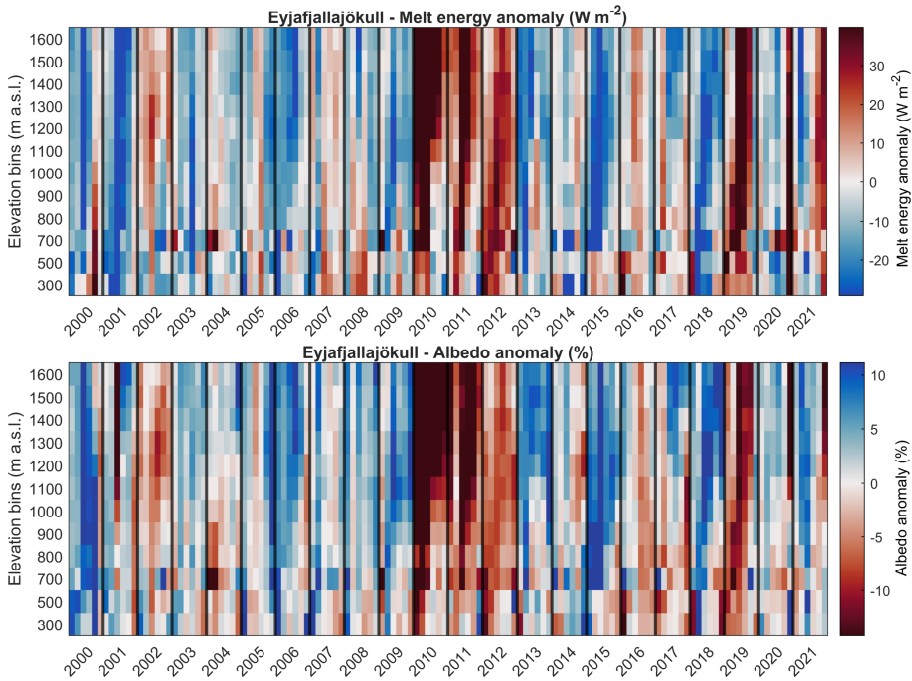

**Figure B4.** Distribution of melt energy and albedo anomalies (W m$^{-2}$) with elevation for Eyjafjallajökull. Vertical axis show elevation bins in 100 m intervals and horizontal axis shows monthly data for each year from April to September. Black vertical lines separate.

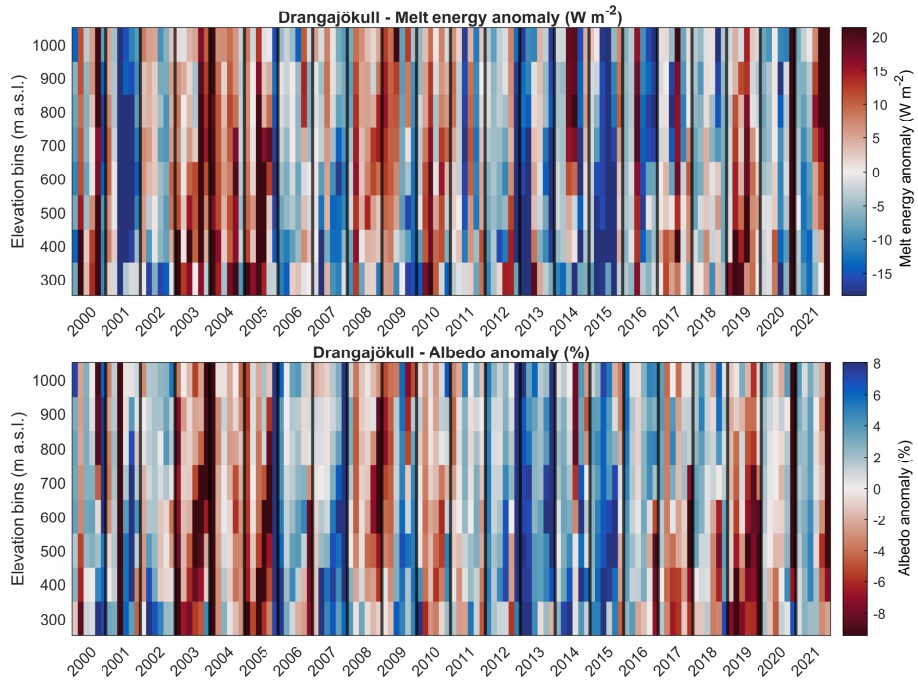

**Figure B5.** Distribution of melt energy and albedo anomalies (W m$^{-2}$) with elevation for Drangajökull. Vertical show elevation bins in 100 m intervals and horizontal axis shows monthly data for each year from April to September. Black vertical lines separate

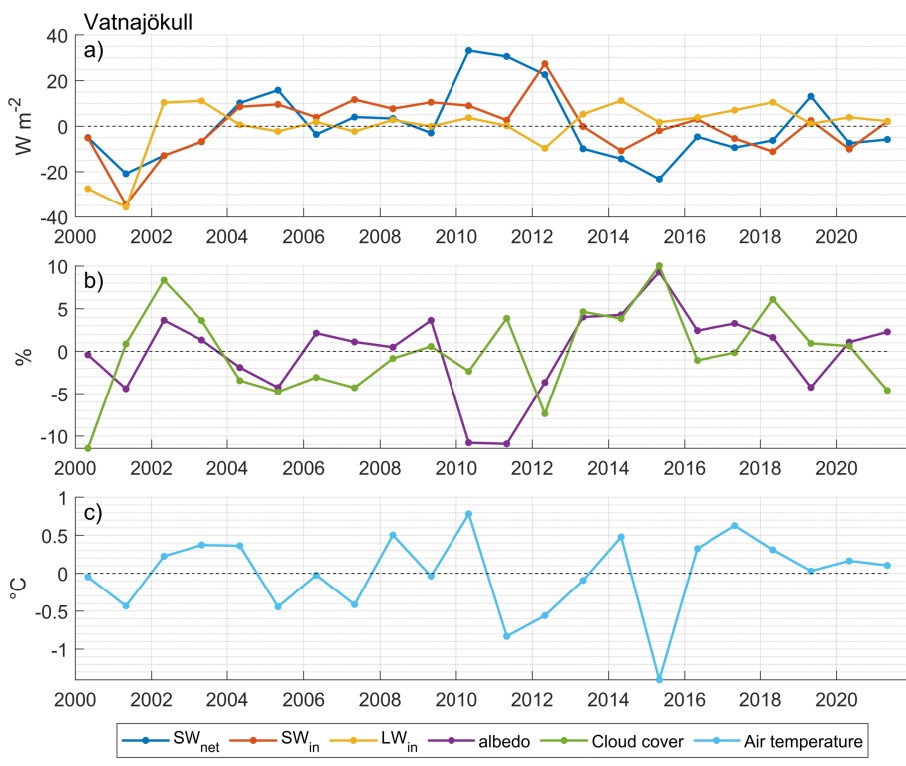

**Figure B6.** Annual anomalies (MJJA) for selected variables for Vatnajökull. a) net short-wave radiation ($SW_{net}$), incoming short-wave radiation ($SW\downarrow$) and incoming long-wave radiation ($LW\downarrow$), b) albedo and cloud cover and c) 2 m air temperature.)

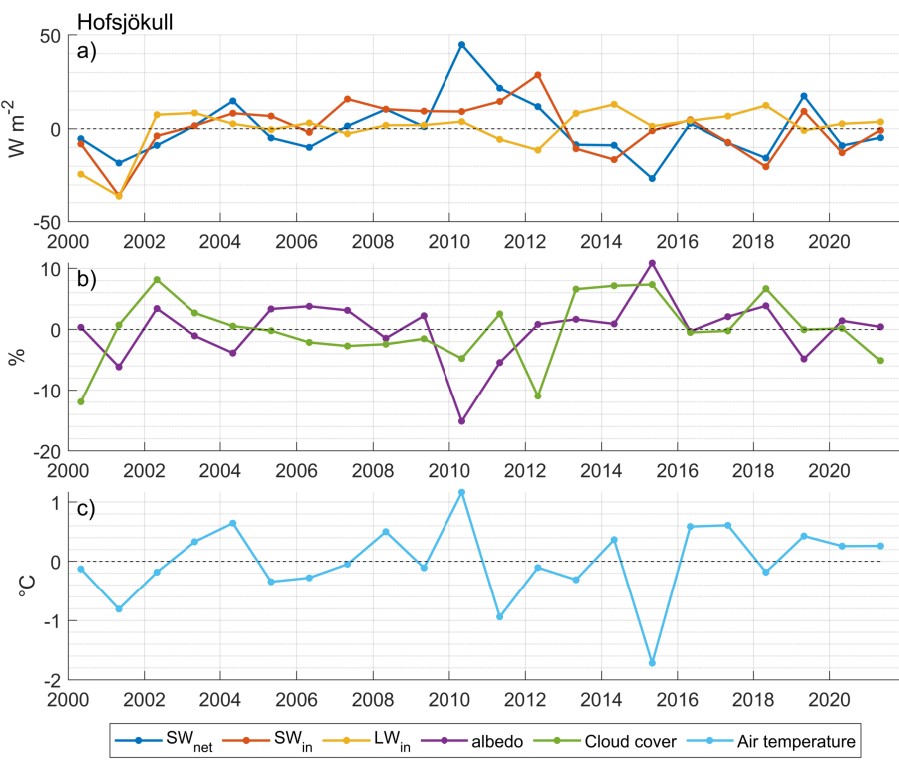

**Figure B7.** Annual anomalies (MJJA) for selected variables for Hofsjökull. a) net short-wave radiation ($SW_{net}$), incoming short-wave radiation ($SW \downarrow$) and incoming long-wave radiation ($LW \downarrow$), b) albedo and cloud cover and c) 2 m air temperature.)

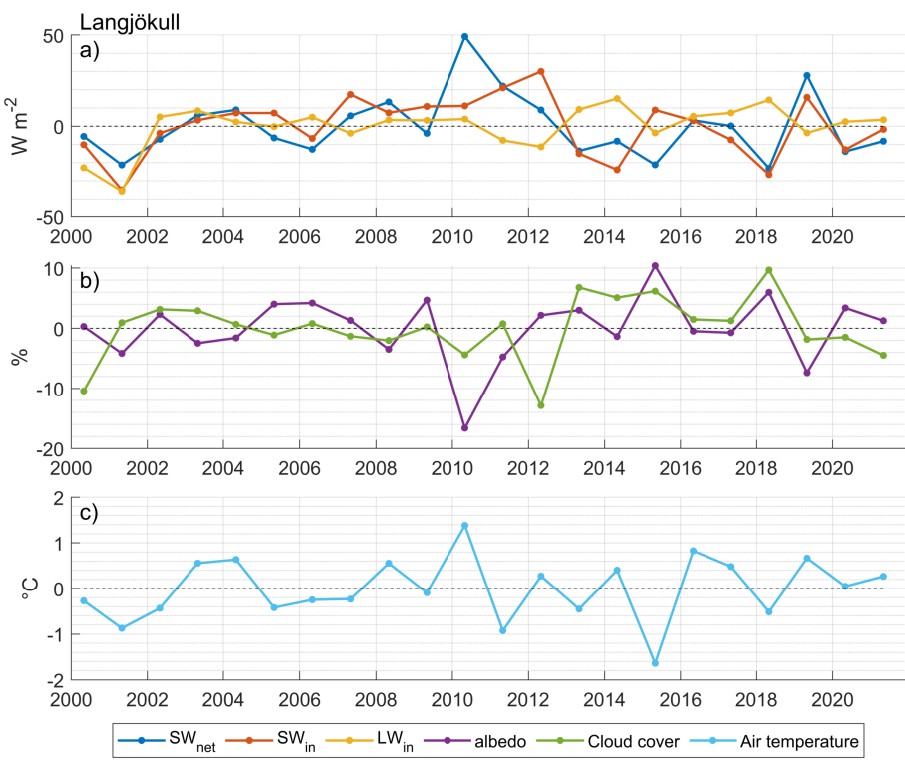

**Figure B8.** Annual anomalies (MJJA) for selected variables for Langjökull. a) net short-wave radiation ($SW_{net}$), incoming short-wave radiation ($SW \downarrow$) and incoming long-wave radiation ($LW \downarrow$), b) albedo and cloud cover and c) 2 m air temperature.)

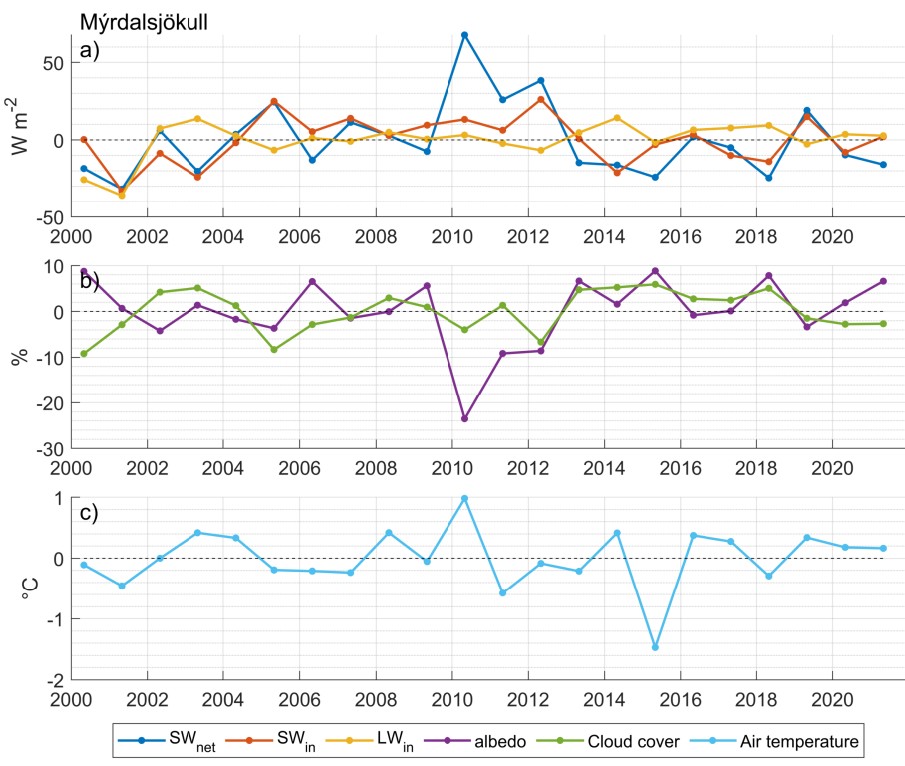

**Figure B9.** Annual anomalies (MJJA) for selected variables for Mýrdalsjökull. a) net short-wave radiation ($SW_{net}$), incoming short-wave radiation ($SW\downarrow$) and incoming long-wave radiation ($LW\downarrow$), b) albedo and cloud cover and c) 2 m air temperature.)

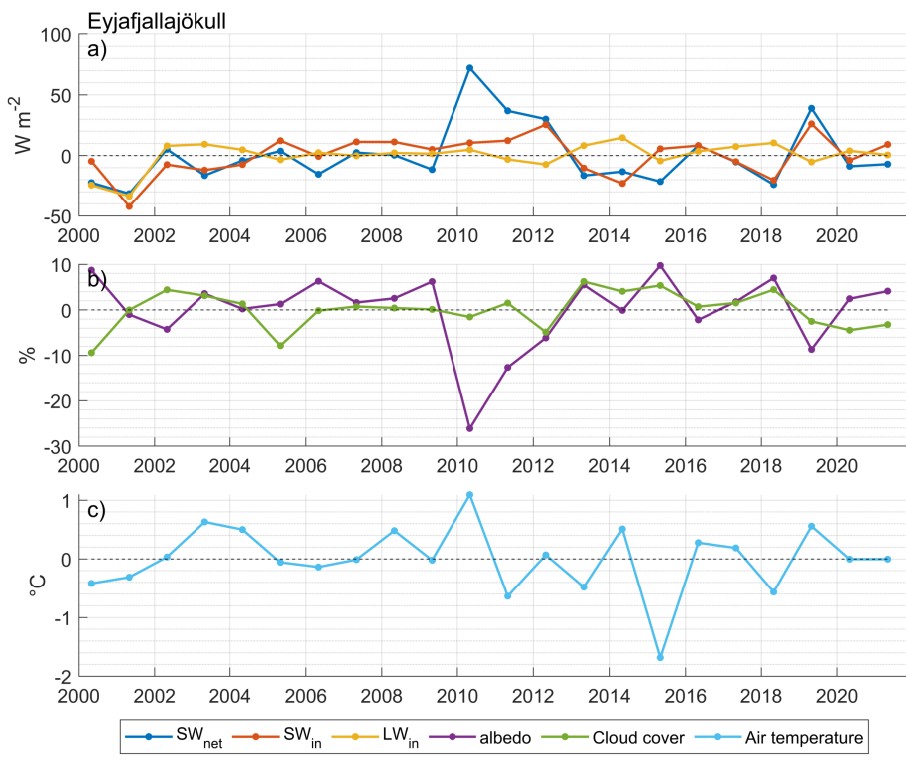

**Figure B10.** Annual anomalies (MJJA) for selected variables for Eyjafjallajökull. a) net short-wave radiation ($SW_{net}$), incoming short-wave radiation ($SW\downarrow$) and incoming long-wave radiation ($LW\downarrow$), b) albedo and cloud cover and c) 2 m air temperature.)

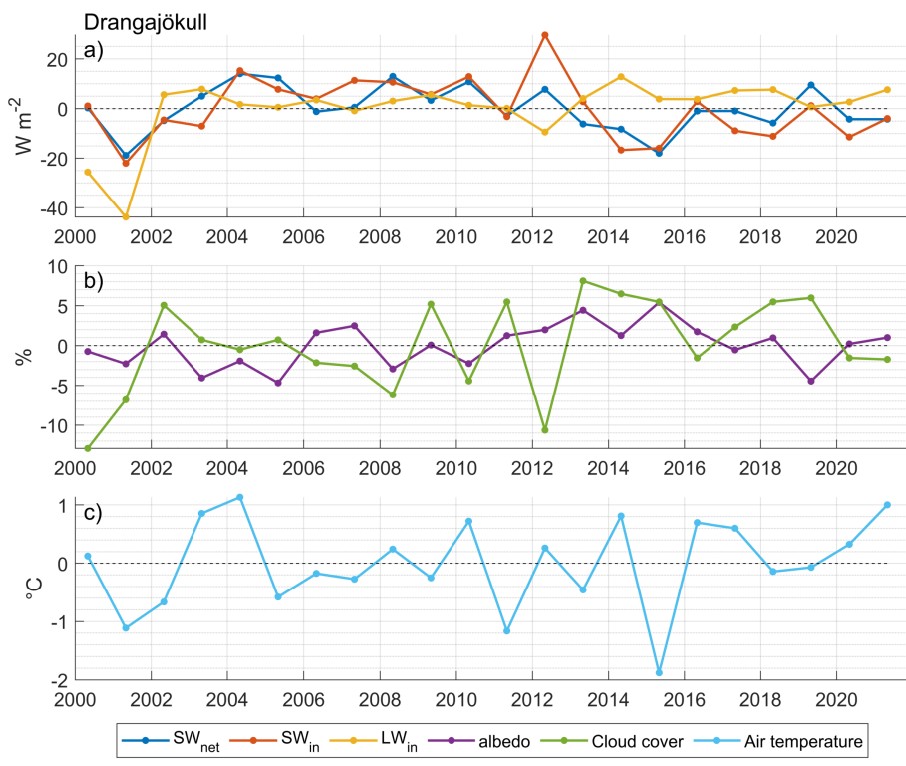

**Figure B11.** Annual anomalies (MJJA) for selected variables for Drangajökull. a) net short-wave radiation ($SW_{net}$), incoming short-wave radiation ($SW \downarrow$) and incoming long-wave radiation ($LW \downarrow$), b) albedo and cloud cover and c) 2 m air temperature.)

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
