# Peer review of "Modeling of surface energy balance for Icelandic glaciers using remote sensing albedo"

_EGUsphere, 2022_

## Author Response (AR1)

**RC1:** 'Comment on egusphere-2022-1088', Anonymous Referee #1, 27 Dec 2022

Reviewer comments in **black**
Author response in **green – submitted 28 Mar 2023**
Author modifications in **red – submitted 09.06.2023**

**General comments**

**RC1-1:** The study provides a detailed look at the energy balance of Icelandic glaciers, with a particular focus on the effect of albedo during LAP events. The authors use an energy balance model and a high-resolution forcing dataset to simulate the melt energy of 6 Icelandic ice caps over the summer. Albedo observations from MODIS were used as model input to decrease the uncertainties associated with this important energy balance variable. The study is generally clear, and the results are described in detail, but could benefit from some minor additions/changes, as outlined below.

**Author response RC1-1:**

We appreciate the thorough and supportive comments to our manuscript, comments are relevant and will improve the work. Please see our answers and suggested modifications below, in **green** the original response and in **red** the responses and modifications to the manuscript.

**Specific comments**

**RC1-2:** L 2-3: It is a bit misleading to write you "developed" an energy balance model, when you use an already existing the wording to "used" or "applied".

**Author response RC1-2:** Agreed, changed accordingly to "applied"

**Author modification RC1-2:** Changed, see Line 2

**RC1-3:** L 30-43: this section mostly describes the study area, so consider moving this part to the "Study area" section

**Author response RC1-3:** This provides an overview of Iceland, climatic boundaries, and drivers of variability. We feel it fits well in the introduction whereas the study area section details in on the glaciers studied in more detail.

**Author modification RC1-3:** No physical change to manuscript.

**RC1-4:** L103-108: I would try to stress the novelty of your work more in this section and the reason for your study. As I understand it, the two major novelties are:

a. other studies have investigated the energy balance of Icelandic glaciers, but these normally only focus on one ice cap or glacier. In this study, you provide a larger context on how the energy balance of Icelandic glaciers have changed. You do mention this in your introduction, but I would stress more that this is often not done.

**Author response RC1-4a:** Agreed – it is important to highlight this better. We add a text addressing this when we address RC1-4b comment, see author response below.

**Author modification RC1-4a:** See response RC1-4b

b. You use remote sensing albedo, which removes one of the major uncertainties that have previously persisted in distributed energy balance studies, as albedo is a hugely important factor for the energy balance in Iceland. Particularly that you can include the lower albedo after dust storms and eruptions is a major plus here. I would stress this more as a purpose of the study.

**Author response RC1-4b:**

Good comment, reviewer 2 had similar comments. We will be modifying L105-108 from:

*The primary objective of this study is to understand and quantify melt season SEB for Icelandic glaciers using high-resolution meteorological climate forcing and remotely sensed glacier surface albedo from the Moderate Resolution Imaging Spectroradiometer (MODIS) sensor. This adds to the previous understanding of spatial and temporal distributions of melt energy, main melt energy sources, and variability within and between glaciers in Iceland and provides insight into the melt enhancement due to volcanic eruptions and years with extensive LAP deposits. It also provides a comprehensive overview of the SEB of Icelandic glaciers since it is not limited to one glacier or glacier outlet.*

To:

*The primary objective of this study is to understand and quantify melt season SEB for Icelandic glaciers using high-resolution meteorological climate forcing and remotely sensed glacier surface albedo from the Moderate Resolution Imaging Spectroradiometer (MODIS) sensor. The study builds on processing pipelines for albedo developed in Gunnarsson et al. (2021). This adds to the previous understanding of spatial and temporal distributions of melt energy, main melt energy sources, and variability within and between glaciers in Iceland and provides insight into the melt enhancement due to volcanic eruptions and years with extensive LAP deposits. In the case of future volcanic eruptions or extensive LAP events, the presented methodology allows for rapid assessment of glacier albedo changes in near-real time and the associated influence on surface energy balance, which can have a direct impact on hydropower production in Iceland and possibly civil infrastructure in some cases. Understanding LAPs processes and impacts on SEB also aids in parametrizations of albedo for other modelling work where remotely sensed albedo may not be available, such as for historic and future modelling studies. The study also provides a comprehensive overview of the SEB of Icelandic glaciers since it is not limited to one glacier or glacier outlet as many previous studies of surface energy balance.*

**Author modification RC1-4b:** See text in Line 79 to Line 85

**RC1-5:** L178-200: Why did you not calculate the local lapse rate from the forcing data? If you have the elevation in each grid point, you could probably calculate monthly lapse rates for each ice cap for all used forcing variables.

**Author response RC1-5:** We decided to apply an external lapse rate from individual studies in Iceland enabling the potential use of lower spatial resolution data (ERA5 or CMIP, for example) where lapse rates from the forcing data would likely be of much poorer quality. Also, if in future work the glacier geometries where to evolve (elevation changes), we could make the relevant lapse rate adjustments annually in the model.

**Author modification RC1-5:** No physical change to manuscript

**RC1-6:** L 208: I am missing some discussion later in the text about the uncertainty of setting the ground heat flux to 0. I know that Icelandic glaciers are temperate, but surely there is a seasonal cold wave that needs to be heated to melting temperature in spring, and thus not all energy can be assumed to be melt energy?

**Author response RC1-6:**

This indeed is a source of uncertainty. Since we do not model the snowpack accumulation during winter the cold content is not easily tracked. To estimate in the model a vertical temperature distribution in the near-surface snow layers would be needed and the thickness/density of the snowpack. Observed temperature in spring mass balance cores indicate that cold content is not a major energy source, see figure below[1], although it provides modulation of the energy.

[Figure]

[1] https://skemman.is/bitstream/1946/35967/1/BS_Verkefni_skil_2020_GBH_Final.pdf

We will add an **Uncertainty sources** section where this would be discussed further.

**Author modification RC1-6:** Discussion incorporated into Uncertainty sources and limitations section, see Line 560 to Line 567.

**RC1-7:** Section 4: In this section, you only validate the forcing data against observations, but not the results of the model. Could you add a validation of the outgoing longwave and shortwave radiation (which should be available from some of the AWSs) and perhaps also the turbulent fluxes?

**Author response RC1-7:**

For selected sites where we have LW and SW outgoing we will add this to the summary tables in the appendix and provide a brief discussion in the relevant section.

Care must also be taken when comparing SW out from the model and AWSs, especially in the bare-ice areas where comparisons to 500 m pixels and stations footprints might not be a fair comparison. Studies from Gunnarsson et al 2021 and Gascoin et al 2017 support this which are cited in the paper.  Turbulent fluxes are not calculated for the AWSs.

**Author modification RC1-7:** Tables added with SW and LW outgoing and RH. Text added in Section 4.1, Line 266 to Line 274. Table A2 has also been added with relevant summary of data.

**RC1-8:** L384-394: consider moving the info about the different eruptions that occurred during the study period to either the introduction or study areas section, as I think the paper would be clearer if this is presented early on (since you mention the eruptions in earlier sections too).

**Author response RC1-8:** L384 to L394 moved to the Study area section and incorporated into the text there.

**Author modification RC1-8:** Modified, see Line 97 to 109.

**RC1-9:** L488: Could you add an "uncertainties" section where you discuss your results? What simplifications have been made, how can other energy balance components be affected by LAP events (both the turbulent and longwave heat flux must change somewhat) etc.

**Author response RC1-9:** Good comment, we will add a section, **Uncertainty sources,** discussing the major assumptions and how they impact the results presented.

**Author modification RC1-9:** Sections *Summary and discussion*  and *Uncertainty sources and limitations* added with relevant discussion.

**RC1-10:** L497: change "eeither" to "either"

**Author response RC1-10:** typo, fixed

**Author modification RC1-10:** fixed, see Line 587

**RC1-11:** Figure 3: you write in the caption that the color scale varies between months, but would it not be possible to use the same scale? It would make comparison much easier between the months.

**Author response RC1-11:** This could be done but the tradeoff is that the individual month distribution and energy dynamics will be lost. We have tested both versions and the current versions was thought to represent the results better.

**Author modification RC1-11:** No physical change to manuscript

**RC1-12:** Figure 4: Could you make the vertical scale the same for all columns? Then it would be easier to compare the different glaciers.

**Author response RC1-12:** This could be done but the tradeoff would be a lot of white space for 5 out of 6 glaciers since the vertical axis would range from 0 – 2000. In an already "tight" layout this would unlikely improve reading the graph.

**Author modification RC1-12:** No physical change to manuscript

**RC1-13:** Figure 5: change "Vatnajokull" to "Vatnajökull" in figure titles

**Author response RC1-13:** fixed

**Author modification RC1-13:** Figure 5 titles have been updated with Icelandic letters and text changed from W/m2 to W m $^{-2}$ to be consistent with The Cryosphere guidelines.

**RC1-14:** Figure 7: could you make the y-axis the same for all figures, so it is easier to compare?

**Author response RC1-14:** Yes, done.

**Author modification RC1-14:** Y-axis limited to -100 to 250 in all panels

**RC1-15:** Figure 8: The text on the figure is too small, particularly on the color bar.

**Author response RC1-15:** Agree, we will make years and color bar larger.

**Author modification RC1-15:** Figure 8 has been modified with larger text, both for the panels denoting which year each figure applies to but also the colorbar. Colorbar text changed from W/m2 to W m $^{-2}$ to be consistent with The Cryosphere guidelines.

**RC1-16:** Figure 8: I find this figure interesting, as the different ice caps mostly follow a similar trend (years with high EB is the same for all ice caps, and vice versa) but there are some noticeable exceptions. Some of this is probably due to ash deposits from eruptions, but e.g. in 2002 and 2003, Mýrdalsjökull and Eyjafjallajökull seem to behave differently that the others ice caps, with a high energy balance in 2002 while the other ice caps have a low energy balance, and the other way around in 2003. 2014 and 2016 also seem to have some ice caps with general high energy balance while others have lower than usual. Is this difference due to dust storms or something else?

**Author response RC1-16:** Due to the proximity to the coast and their lower overall elevation Eyjafjalla- and Mýrdalsjökull can experience differences in climate during the melt season.

Spring and early summer fresh snow fall are not uncommon, especially at Mýrdalsjökull that would lower the SEB for the season. They are also surrounded by erosive LAP dust hotspots that can contribute LAPs that other glaciers do not experience from areas such as Mælifellssandur, Mýrdalssandur and Markarfljótsaura. These areas are technically in the lowlands of Iceland and are exposed much earlier underneath the seasonal snow in Spring or late winter and in some cases do not sustain seasonal snow during winter. In 2002 the SEB is higher but also at the snout of SW Vatnajökull (Síðujökull outlet) which could originate from the same dust hotspot. Although not reported in this study, during early spring 2022 Mýrdalsjökull, Eyjafjallajökull and other smaller glaciers near the south coast experienced a LAP event not effecting the other larger ice caps. Discriminating between the impacts dust has on albedo and rain/natural snow metamorphosis is challenging only from remote albedo but remains a very interesting topic to study to further understand the impacts LAPs have on SEB. It has been done in many locations, but we have not had the time to focus on it yet.

L365 mentions this "The south-coast glaciers were also close to unstable dust hotspot areas where seasonal snow melts out earlier than in the highlands, exposing erosive surfaces."

**Author modification RC1-16**: No physical change to manuscript

**RC1-17:** Figure B1-B5: change "jokull" to "jökull" and "Myrdals" to ""Mýrdals".

**Author response RC1-17:** Yes, this will be done.

**Author modification RC1-17**: Figure B1-B5 titles have been updated with Icelandic letters and text changed from W/m2 to W m $^{-2}$ to be consistent with The Cryosphere guidelines.

**RC2:** ['Comment on egusphere-2022-1088'](), **Anonymous Referee #2, 31 Jan 2023**

Reviewer comments in **black**
Author response in **green – submitted 28 Mar 2023**
Author modifications in **red – submitted 09.06.2023**

**General comments**

**RC2-1:** The authors of the study examine the surface energy balance of the largest Icelandic glaciers from 2000 to 2021, taking into account the ablation period from April to September. They apply an energy balance estimation using MODIS-derived albedos. High-resolution WRF data with an hourly resolution of 2 km are used as meteorological forcing. The results show the large spatial and temporal variability of the melting energy. The energy balance terms are presented in detail, resolving the different glacier areas, elevations and seasonal and annual patterns. The special feature of the study, in my view, is the coverage of almost the entire ice-covered area (97%) of Iceland and especially the focus on the influence of light-absorbing particles on the energy balance. These particles come from sand deserts and volcanic eruptions.

The paper is excellently written, and the text is easy to understand. It was enjoyable to read the manuscript.

In my opinion, the publication can make a valuable contribution to the knowledge of glaciers in Iceland. However, I have some points that should be (better) addressed before publication. I hope that the following major, specific and technical comments can help improve the manuscript.

**Author response RC2-1:**

We appreciate the thorough and supportive comments to our manuscript. It is clear that the reviewer spent a lot of time reviewing the manuscript in great detail which is greatly appreciated. Please see our answers and modifications below, in **green** the original response and in **red** the responses and modifications to the manuscript.

**Major comments**

1. **RC2-2: Novelty and differentiation from Gunnarsson et al. (2021):** I like the idea of building on a previous study using the derived gap-filled and post-processed MODIS dataset. My main issue is that some of the results and conclusions are similar to those of the 2021 TC paper. You mentioned several times that SWnet is modulated by the albedo and that the melt patterns are mainly driven by SWnet. Therefore, it is obvious that e.g. the elevation gradient of the albedo in Gunnarsson et al. (2021) is consistent with the elevation gradients of the melt energy presented here. Therefore, at the end of the introduction, I would recommend clearly elaborating what the similarities and differences are to Gunnarsson et al. (2021), how this study extends Gunnarsson et al. (2021) and what makes this study unique.

**Author response RC2-2:** We will add sentence in the final paragraph in the introduction sentence that details how this study builds on work from Gunnarsson 2021 and emphasize the novelty of the study. We do mention many of the similarities, as expected, to Gunnarsson et al 2021, e.g., L330 "Albedo gradients from Gunnarsson et al. (2021) follow similar patterns with elevation to those of SWnet (general albedo increase with elevation) for all the glaciers, demonstrating how SWnet was modulated by albedo."

Modifications of the objectives in the final paragraph in the introduction have been modified accordingly:

*The primary objective of this study is to understand and quantify melt season SEB for Icelandic glaciers using high-resolution meteorological climate forcing and remotely sensed glacier surface albedo from the Moderate Resolution Imaging Spectroradiometer (MODIS) sensor. The study builds on processing pipelines for albedo developed in Gunnarsson et al. (2021). This adds to the previous understanding of spatial and temporal distributions of melt energy, main melt energy sources, and variability within and between glaciers in Iceland and provides insight into the melt enhancement due to volcanic eruptions and years with extensive LAP deposits. In the case of future volcanic eruptions or extensive LAP events, the presented methodology allows for rapid assessment of glacier albedo changes in near-real time and the associated influence on surface energy balance, which can have a direct impact on hydropower production in Iceland and possibly civil infrastructure in some cases. Understanding LAPs processes and impacts on SEB also aids in parametrizations of albedo for other modelling work where remotely sensed albedo may not be available, such as for historic and future modelling studies. The study also provides a comprehensive overview of the SEB of Icelandic glaciers since it is not limited to one glacier or glacier outlet as many previous studies of surface energy balance.*

**Author modification RC2-2:** Changed, see Line 75 to 85

2. **RC2-3: Energy balance: wind speed? surface temperature measured or iteratively solved? humidity at the surface?:** The explanations regarding the energy balance estimates are not detailed enough in my opinion. According to Section 3.3, air temperature, surface temperature, incoming long and short wave radiation, barometric pressure and specific humidity are used. The wind speed which is of major importance for the turbulent fluxes, is missing. You could use the output wind fields from WRF. According to equation (5) wind speed is needed? Furthermore, what is used for the surface humidity at height z0? The WRF output or is the surface humidity assumed to be 100 %? Another question I have is the surface temperature. Is the surface temperature used from WRF? According to the text the SEB was solved iteratively for surface temperature (Line 205), but according to Section 3.3, the surface temperature is used from WRF.

   **Author response RC2-3:** Wind speed is indeed used from WRF and surface temperature is not. We will modify sentence L170-L172 accordingly:

**From:** Relevant meteorological surface data were extracted for use in the energy balance model, including air temperature at 2 m, surface temperature, incoming long- and short-wave radiation, barometric pressure at surface level, and specific humidity; all were resampled to daily average values.

**To:** Relevant meteorological surface data were extracted for use in the energy balance model, including air temperature at 2 m, wind speed, incoming long- and short-wave radiation, barometric pressure at surface level, and specific humidity; all were resampled to daily average values.

Surface humidity at height z0 is evaluated by the model. L230 provides reference to this "Surface roughness lengths for heat and moisture were calculated for snow and ice separately as in Van As (2011)". More details are in Van As (2011) and the model code is found in Vandecrux, 2018.

**Author modification RC2-3: Changed, see Line 160 - 162**

3. **RC2-4: Uncertainties, simplifications and limitations:** I think you are aware of the uncertainties, simplifications and limitations of the study. Nevertheless, in my opinion, these are too little discussed in the present study. One idea would be to add a subsection after the validation and collect and discuss the different issues. If you add a section before presenting and discussing the results you directly show that you are aware of these issues. In the following I will highlight only some of the things I thought of. Sometimes you have mentioned the points I thought of when reading the manuscript, but only at a later stage.

   o If you have a tephra layer above the snow and ice surface you can have surface temperatures > 273.15 K in reality. **(see Section 5 and 5.1)**

   o If I understand correctly, you calculate the albedo from an 11 day average. So if there is a thin snow cover from a summer snowfall lasting only 2 days, for example, this will be underrated in your approach. It is not essential for the results but I think you still can mention such limitations. **(see Section 5 and 5.1)**

   o There are limitations within the WRF data. Line 162: I would recommend at least discussing the uncertainties when you combine WRF datasets with a totally different forcing (ERA-Interim versus NCEP). In my opinion your combined forcing dataset is not consistent anymore. It is ok to use the combined dataset but this issue has to be mentioned and discussed. **(see Section 5 and 5.1)**

   o Your SEB estimation has limitations and simplifications. Can you name more effects and discuss them including references? E.g. the bulk approach, LWout parameterisations. **(see Section 5 and 5.1)**

   o In the 2021 TC paper you write: "[...] Vatnajökull, and boundaries in 2007 and 2008 were used for Langjökull and Hofsjökull, respectively. This was selected as a midpoint representing an average glacier area during the

period 2000–2019. This needs to be considered when interpreting rapid changes at the glacier terminus, as some areas in 2000 were part of an active glacier but might in 2019 be dead ice or land." and for example "It is important to consider how representative point-based in situ observations are (observing ~ 120–180 m2; Kipp and Zonen, 2019), compared with the spatial footprint of the MODIS data (0.25 km2), especially in glaciated areas with high spatial albedo variability and MODIS sub-pixel variability as is observed in the bare-ice areas of the Icelandic glaciers." I think such considerations should be made here as well, adapted to the used data.

**(See Line 97-99 and Line 525-526, Line 573-577 )**

- o  Line 257 a temperature bias up to 1.15 K. I suggest removing slightly, and discuss the uncertainty resulting from the bias. 1 K makes a difference.

  Sligthly removed, see Section 5 and 5.1

**Author response RC2-4:** Agreed, valid suggestions and points relating to the uncertainty of the model. We will add an **Uncertainty sources** section in the results where this would be discussed further and in some cases in the results text where applicable.

**Author modification RC2-4:** Changed, see sections and line numbers referred to above.

4. **RC2-5: Validation:** Some of the meteorological forcing variables are validated. What about barometric pressure, specific humidity and probably wind speed. How did you downscale the barometric pressure from the 2 km WRF grid to the 463 m MODIS grid? The MODIS data are validated in the 2021 TC paper. That's good. But I could not find a validation of the calculated energy balance terms (SWin, LWin, LWout, SHF, LHF) and the resulting potential melt energy? In my opinion the validation of the results could be done within the discussion of the results or in a specific subsection of the results and discussion section. I understand that there may not be directly comparable data. But you could use other studies in Iceland on single glaciers, or use studies on the Greenland ice sheet, or in Svalbard, or in Scandinavia to at least assess the range of the calculated values. Furthermore, you could also compare relative values with Björnsson (1971, 1972), and more recent studies. In the abstract you write: "Validation was performed using observations from various glaciers spanning distinct locations and elevations with good visual and statistical agreement." after the sentence: " The SEB was reconstructed from April through September for 2000–2021 at a daily timestep with a 500 m spatial resolution." So I expected a statistical validation of the SEB.

**Author response RC2-5:** Agreed, our focus is mostly on the radiative components and temperature providing validation to the main drivers/forcings of SEB in the model. As pointed out albedo has been validated at the AWSs sites in Gunnarsson et al 2021.

Barometric pressure among other variables except for air temperature and long-wave radiation were downscaled using bi-cubic interpolation between the grids. We will add a sentence in Section 3.3 clearing this up.

**Author modification RC2-5a:** Added in Line 168-169: *"Other meteorological forcing data was downscaled with bi-cubic interpolation."*

Indeed, validation of SWin, Lwin is provided in Section 4.1 which are energy balance components. Validation of turbulent fluxes is challenging; they are note calculated for the AWSs data as in many cases certain observations to calculate EB are missing. This would need gap filling either from neighboring AWSs on land or from WRF data, introducing more uncertainties or in a sense a completely separate study.

For selected sites, we have LW outgoing, SW outgoing and relative humidity and could add this to the summary tables in the appendix and provide a brief discussion in the relevant section in text. Wind speed is also observed but generally at 3-5 m height above the glacier surface meaning that for an accurate comparison downscaling from the WRF 10m elevation is needed.

**Author modification RC2-5b:** We have added comparison to LW and SW outgoing and relative humidity which was overall available. Tables added with SW and LW outgoing and RH. Text added in Section 4.1, Line 266 to Line 274. Table A2 has also been added with relevant summary of data.

We do provide reference to Schmidt et al., 2017 in many locations both with respect to the validation (e.g., Section 4.1) and in the results in general. Note should be taken that there are note multiple studies of energy balance of Icelandic glaciers. The most recent work by Schmidt et al., 2017 is used as a "benchmark" study. Comparison to Björnsson (1971, 1972) is challenging as it focuses on small alpine glaciers that originate in high mountains in small bowls with steep sides (cirques).

L272 to L280 do provide some comparison to other validations in Iceland and Greenland. Many of the limitations due to calculations of turbulent fluxes are discussed in Section 3.4.

We will move the validation section into the Results main chapter and provide validation details on further comparison to other studies there.

**Author modification RC2-5c:** Changed, Validation of meteorological forcings is now a subsection within the Results chapter.

5. **RC2-6: Estimated SW radiative forcing from LAPs:** Line 443–445: If I understand correctly, you use the same climate forcing e.g. for the year 2010, first with the mean albedo (2000–2021), then with the observed albedo in 2010. Besides LAPs the observed albedo in 2010 could also be influenced by climate, or? In Line 481 you state that for example snowfall has an impact on SWfLAP,. Please explain your setup

in detail and discuss this issue. I like the approach and the investigation, it would just be good if you could show that you are aware of the limitations and possible influence of e.g. snowfall and temperatures on the observed albedo in certain years.

**Author response RC2-6:** We provide details on the methodology in L442 – L452 and mention the limitations of the approach. In L449 it says: "This approach does not fully consider all physical processes: e.g., as it did not take into account the effect on albedo of different snow metamorphosis processes between years, or the timing of melt-out of impurity-rich ice; but in this comparison these processes were secondary to the overwhelming impact LAPs had on the albedo, especially in 2010 and 2011. Additionally, the impacts on turbulent fluxes were ignored as they are considered negligible."

And in L486 it says, "Limited data were available to fully estimate where isolation might have occurred, and more complex modeling is needed to fully represent the problem."

These effects and limitations to the approach are indeed mentioned. In the added Uncertainty section that will be added we will expand the discussion on these effects.

**Author modification RC2-6:** In addition the above response we have added the Uncertainty sources and limitations section with further discussion of these impacts.

**Specific comments**

**RC2-7: Titel:** The title 'Modeling of surface energy balance' is very general. Perhaps sth. about LAPs, volcanic impacts, .. could be added.

**Author response RC2-7:** The title is general as it provides overview information about SEB of Icelandic glaciers. Adding to the title would results in a very long title, that already is quite long.

**Author modification RC2-7:** No change

**RC2-8: Abstract:** There are no numbers from your results in the abstract. Maybe the mean melt enhancement (in %) from LAPs could be added to the abstract.

**Author response RC2-7:** Good point, a sentence quantifying the melt enhancement from LAPs for 2010, 2011 and 2019 will be added to the abstract.

**Author modification RC2-8:** Added in the abstract: *The impact of LAPs was often significant even though the glaciers were far away from the eruption location. On average, melt enhancements due to LAPs were ∼ 27% in 2010, ∼ 16% in 2011 and ∼ 14% in 2019, for Vatnajökull, Hofsjökull and Langjökull.*

**RC2-9: Line 7:** What is the difference between annual variability and inter-annual variability? By annual variability, do you mean intra-annual variability or seasonal variability? For me, annual variability is the same as inter-annual variability. You wrote seasonal and inter-annual in the heading of section 5.1 and in the conclusion Line 493: "[...] melt-season and inter-annual variability [...]".

**Author response RC2-9:** We will streamline this throughout the paper and consistently use inter-annual and seasonal variability. observed in different glaciological years is what we are aiming at. Seasonal refers to within the season, April through September.

**Author modification RC2-9:** Line 583 in conclusion changed to: "***The main results show large seasonal and inter-annual variability…***"

**RC2-10: Line 32:** Could you explain a little more in depth what " high precipitation sustaining a seasonal snow pack and glaciers" means. In which month/season is the precipitation peak? Which months are the driest?

**Author response RC2-10:**

**Author modification RC2-10:** Line 33: *Iceland has a maritime climate with mild winters, cool summers and high average precipitation, especially in the fall and winter, sustaining a seasonal snow pack and glaciers*

**RC2-11: Introduction:** The introduction is rather long with 1650 words. Please revise this section and check which sentences are really needed for the motivation of the study. The historical background in Line 73–102 is very interesting. Nevertheless, I think these paragraphs can be shortened.

**Author response RC2-11**: Agreed, we will make modifications aimed a shortening or removing the introduction with focus on L73 – 102

**Author modification RC2-11:** Suggested text removed to shorten the introduction section.

**RC2-12: Methods:** In contrast to the introduction the Methods section is rather short, especially the presentation of the surface energy balance and the parameterization of the different terms (cf. Major comment 2). Is there a storage (snow/ice temperature) which is not mentioned or how is the cold content from winter (Line 378-379) resolved by the estimation. The sub-surface heat flux which could transport cold content to the surface is assumed to be zero (Line 209).

**Author response RC2-12**: Cold content of the snow is assumed to be zero along with energy from precipitation. This indeed is a source of uncertainty. Since we do not model the snowpack accumulation during winter the cold content is not easily tracked. To estimate in the model a vertical temperature distribution in the near-surface snow layers would be needed and the thickness/density of the snowpack. Observed temperature in spring mass

balance cores indicate that cold content is not a major energy source, see figure below[2], although it provides modulation of the energy.

[Figure]

We will add an **Uncertainty sources** section where this would be discussed further.

**Author modification RC2-12:** Discussion incorporated into Uncertainty sources and limitations section, see Line 560 to Line 567.

**RC2-13: Line 132–132**: Did you derive the albedo again from the MODIS product? Or did you use the dataset from Gunnarsson et al. (2021). If you used the dataset, I recommend rephrasing the sentence accordingly.

**Author response RC2-13**: We did update the work from Gunnarsson 2021 with version 6.1 from 6.0 of MOD10A1 and MYD10A1 MODIS products. The processing pipeline remained the same. We will rephrase the sentence accordingly.

**Author modification RC2-13:** Added Line 128-129: The albedo data produced in Gunnarsson et al. (2021) was based on version 6.0 of the MODIS data but in this study reprocessed using version 6.1 without modifications to the processing steps.

**RC2-14: Line 140–150:** The final used albedo product has a daily resolution or? I would name the final used temporal resolution in this paragraph. I was first confused with the 11 days buffer.
* * *
[2] https://skemman.is/bitstream/1946/35967/1/BS_Verkefni_skil_2020_GBH_Final.pdf

**Author response RC2-14**: Yes, the final version as a daily resolution.

We will do the following modification L147 (XX):

From: …the mean was calculated to represent the surface albedo, after median…

To: …the mean was calculated to represent the daily surface albedo used, after median…

**Author modification RC2-14:** See Line 137

**RC2-15: Line 176–177**: Did you adjust the WRF output to the original IslandsDEM or to the MODIS grid (463 m)?

**Author response RC2-15**: IslandsDEM was regridded to the MODIS grid and then applied with WRF regridding.

**Author modification RC2-15:** see L163-165: "To downscale the meteorological forcing data from the 2 km WRF grid to the 463 m MODIS grid, the model uses the IslandsDEM digital elevation model from the National Land Survey of Iceland (accessed June 1, 2020). A 20 m version of the elevation model was resampled to the native MODIS grid for further processing with bi-cubic interpolation"

**RC2-16: Line 187**: What does "environmental" lapse rate mean in this context?

**Author response RC2-16**: environmental changed to temperature.

**Author modification RC2-16:** See Line 179

**RC2-17: Line 201**: Due to the usage of 5 days backward/forward in case of the MODIS data and the original hourly WRF forcing and three different spatial resolutions it was not directly clear for me what is the final temporal and spatial resolution. Maybe you can add to the sentence in Line 201: "[...] using estimations of daily SEB with a resolution of 463 m." if I am correct. Or sth. similar indicating the final spatio-temporal resolution.

**Author response RC2-17**: We will change L201 from:

The physical processes driving surface melt over snow- and ice-covered surfaces are isolated using estimations of the SEB.

To:

The physical processes driving surface melt over snow- and ice-covered surfaces are isolated using estimations of the SEB at a daily timestep with a spatial resolution of 463 m.

**Author modification RC2-17:** See L194

**RC2-18: Section 4.1:** Is there a reason why you present R2 for T2 and LW but not for SW?

**Author response RC2-18**: We will add R2 values for SW in a similar manner as for T and LW

**Author modification RC2-18:** See L253-254, R2 values have been added.

**RC2-19: Line 249:** Maybe there is a misunderstanding from my part, but for me SW, LW, .. are the energy balance components. So maybe you mean: "The downscaled meteorological forcing [...]" instead of "The downscaled energy balance components [...]".

**Author response RC2-19**: Agreed, will change *energy balance components* to *meteorological forcings*.

**Author modification RC2-19:** Updated, see Line 240

**RC2-20: Results and discussion:** Sometimes it is difficult to recognise which are results of the study and which are results evaluated with the help of other studies. Separating the results and discussion into two different sections would help here. With this, the discussion could also be conducted more independently of the order of the graphs. Furthermore, the discussion could be expanded. Especially the comparison with other studies with numbers would be helpful. This comment adds to the validation of the energy balance terms (major comment 4). The comparison in Line 316–320 is very general and all studies are cited at the end of the paragraph. Readers will be interested in a more in depth comparison of what is similar and what is different. Besides the calculated energy balance terms and the available melt energy the gradients could be compared (Line 310–315). Furthermore, you can discuss that you found positive albedo trends over the study period in northern Vatnajökull in the TC 2021 paper, but no significant trends were found in this study.

**Author response RC2-20:** There are always pros and cons writing the results and discussion together or separately. In this case we did it together as we felt it helped the understanding of the study and the flow in presenting our results. A short discussion section will be added, where the key points of the paper are discussed, implications in understanding SEB variability, how that benefits future work where albedo is not from observations but model parametrizations, the importance of correctly estimating albedo and how extensive the LAP events can be on SEB, influence of LAP deposition timing of the impacts, etc.

**Author modification RC2-20:** Section 5, *Summary and discussion*, has been added

**RC2-21: Line 285:** In my opinion, you cannot see the inter-annual variability with Figure 3. You can see the seasonal and spatial variability. But extreme positive or negative years are not visible.

**Author response RC2-21**: in reference to **Author response RC2-9** we will update inter-annual to seasonal.

**Author modification RC2-21:** Inter-annulal **has been updated to seasonal.** Sentence in L285 (original version line numbers)"Inter-annual SEB variability for Icelandic glaciers was generally high" was moved to L392 (new version line numbers) referring to Figure 8 detailing the inter-annual variability.

**RC2-22: Line 291:** I understand between 10 and 15 % of the mean annual (2000-2021) melt energy was observed. If so, think of adding 'mean annual'.

**Author response RC2-22:** agreed.

**Author modification RC2-22:** Updated, see Line 293

**RC2-23: Line 321–322:** Please add a reference to "other Northern Hemisphere glaciers and ice sheets".

**Author response RC2-23:** Added Hock 2005 and Six, 2009 and modified LXXX

**Author modification RC2-22:** see Line 324

**RC2-24: Line 324–325:** Can you add a short statement how a negative correlation between LWnet and SWnet increases the contribution of the sensible heat fluxes? You mean the relative contribution?

**Author response RC2-24:** Yes, this is not worded well enough. We will rewrite and make the sentence clearer.

**Author modification RC2-24:** This sentence has been modified from:

A negative correlation between LW$_{net}$ and SW$_{net}$ results in net radiation less than SW$_{net}$, increasing the contribution of the sensible heat fluxes to summer melt, especially in the ablation area, less so at higher elevations.

To:

Generally, net radiation contribution was reduced as LW$_{net}$ was mostly an energy sink (negative), reducing the SW$_{net}$ contribution, increasing the relative contribution of the sensible heat fluxes to melt.

See Lines 326 - 328

**RC2-25: Line 328:** How do you know that the albedo was mainly driven by climatology? From the applied method or from another source. I recommend adding a short explanatory sentence or a reference. In the MODIS data you just see the evolution of the albedo, but in the first place you do not see the reason, for example, for a sudden decrease.

**Author response RC2-25:** Agreed, this is not worded well enough and will be rephrased in a more clear way.

Modified from:

Albedo evolution in the accumulation area throughout the melt season was mainly driven by climatology, i.e., snow metamorphosis, not LAPs, although events of sand- and dust deposits could be observed in the albedo data for individual years, impacting SWnet.

to:

General albedo evolution in the accumulation area throughout the melt season was mainly driven by climatology, i.e., snow metamorphosis, not LAPs, although events of sand- and dust deposits could be observed in the albedo data for individual years, impacting SWnet. MODIS albedo data does not allow for discrminating between snow metamorphosis and LAPs impacts, but this assumption was based on albedo data in the accumulation area that seldomly reaches values low enough to reflect annual and extensive LAPs in the surface, unless realated to volcanic eruption years. Figure 6 in \ciptep shows the average elevation distribution of albedo.

**Author modification RC2-25:** Rewritten ,see Line 333 - 336

**RC2-26:  Line 347–348:** Can you add a reference to "warm southerly winds and precipitation" and I guess you mean liquid precipitation or? So maybe add "liquid" to precipitation or change precipitation to "rain".

**Author response RC2-26**:  This is based on monitoring of these events through various met forecasts and observations and the meteorological forcing data used in the study. There is no official publication that we can cite on this but overall, we are using the forcing data to detail this among AWS observations. We will add liquid to precipitation.

**Author modification RC2-26:** Liquid added, see Line 355. Citation added in L357 and 362

**RC2-27: Line 344**: I think you can partly restructure the discussion. Here you are already talking about the impacts of volcanic eruptions before the subsection "5.2 Impacts of volcanic eruptions and other LAP events" starts.

**Author response RC2-27**: Here we are discussing figure 5 making the reference to these events relevant explaining the impacts. The figure shows that in 2010 and 2011, tephra deposits in the upper elevations, from the eruptions in Eyjafjallajokull (2010) and Grimsvotn (2011), greatly enhanced radiative forcing in the accumulation area. We will change *"greatly enhanced radiative forcing in the accumulation area"* to *"greatly impacting albedo in the accumulation area."* In L344

**Author modification RC2-28:** See Line 352

**RC2-28: Line 350**: How do you know that the LAP deposits are from the near pro-glacial areas and not from somewhere else?

**Author response RC2-28**: This is based on monitoring of these events in 2019 through various satellite images and operational web cams in these areas. These is no official publication that we can cite on this jet. We will cite **Unpublished data**.

**Author modification RC2-28:** We have added (unpublished data, based on satellite images and operational web cameras in the field) in Line 357

**RC2-29: Line 352–354:** You probably got the information about "clear skies" and "cold temperatures" from the weather stations. But where does the information about the winds come from? I could not find the information in the manuscript. Please add somewhere a sentence with reference, maybe in the methods.

**Author response RC2-29**: This is based on monitoring of these events through various met forecasts and observations and the meteorological forcing data used in the study. There is no official publication that we can cite on this. We have winds both from observations on glaciers, land and various forecasts and summary reports. We will add a reference to the annual mass balance report that contains a lot of this information and reporting done by the Icelandic Met Office and published online[3].

**Author modification RC2-29:** Reference to the mass balance report in 2020-21 and the annual climate overview from the IMO added as references.  See Line 362

**RC2-30: Line 372**: Do you have an assumption or can you discuss why cloud cover and LWnet were not significantly correlated?

**Author response RC2-30**: This has not been analyzed to detail. Spatial distribution of cloud cover over glaciers in Iceland is reported in Gunnarsson et al 2019 and 2021 which certainly has an impact. More clouds are often observed in the lower bare-ice elevations where land and ice meet. Then higher up in the accumulation zone cloud cover is higher. In between you often have areas extensive in surface area with much lower cloud cover. Even though the relationship is not significant it has quite a good correlation 0.72 for Vatnajökull.

**Author modification RC2-30:** no manuscript modifications

**RC2-31: Line 385:** When you cite explicit numbers, a direct reference would be good. Please add a reference to 0.06 km3. The same applies to Line 390.

**Author response RC2-31**: L385 and L390 has the reference in the following line, we can duplicate them at the end of the sentence.

**Author modification RC2-31:** referenced added to the corresponding lines, see Line 100-109

**RC2-32: Line 442**: "The impacts … 2004, 2010, 2011 and 2019 were assessed." and in Line 444 "observed albedo in 2010, 2011 and 2019." How was 2004 assessed? Using the
* * *
[3] https://vedur.is/vedur/vedurfar/manadayfirlit/2022

observed albedo in 2004 or 2005? Because the event was in fall 2004 if I understood correctly. Please specify.

**Author response RC2-32**:

L454 explains this: "In 2005 the SWfLAP was 4.3 W m−2, here associated with the November 2004 eruption in Grimsvötn"

**Author modification RC2-32:** no manuscript modifications

**RC2-33: Line 443 and Figure 9**: Inconsistent. In the text you write: "mean albedo for the study period (2000-2021)" in the caption you write: "average albedo (2000–2021 mean excluding 2010, 2011 and 2019 in the mean)".

**Author response RC2-33**: Will be fixed.

**Author modification RC2-33: see** Line 443-444

**RC2-34: Conclusion:** One of your conclusions is the influence by high climate variability. To support this statement you could create some monthly (2000–2021) and annual plots of the different forcing variables placed in the appendix. These plots could also support the discussion of the climatic influences on the melt patterns in the "Results and discussion"' section.

**Author response RC2-34**: Will be fixed

**Author modification RC2-34:** Figures have been added to the appendix showing anomalies for incoming short-wave and long-wave radiation, net short-wave radiation, cloud cover and albedo and air temperature. Section ***Summary and conclusion*** refers to them.

**Technical, minor comments**

**RC2-35: Line 30**: I think the dot within 103.000 is wrong. I assume you wanted to use a thousands separator. If so, you should rather use a comma. The same applies to Line 44 and Line 49. Please also check the other sections. If you want to use a thousands separator you should also use it everywhere. For example in Line 44 "3400" and Line 221 "1005".

**Author response RC2-35**: The paper is typeset for US-English. This will be fixed.

**Author modification RC2-34:** 103.000 and 22.000 changed to 103,000 and 22,000. Other sections checked as well. Line 31, 45, 50

**RC2-36: Line 174**: I recommend adding the url to the dataset here: "(https://www.lmi.is/, last access: June 1, 2020)". And maybe you have a link that points specifically to the dataset and not to the main page.

**Author response RC2-36**: Will be update to the dataset location:

**Author modification RC2-36:** https://atlas.lmi.is/mapview/?application=DEM added to the reference in text. Direct link to download the DEM data. Line 164-165

**RC2-37: Line 179 and all further units of Kelvin**: "6–7° K" should be "6–7 K" without the degree sign. Furthermore here you do not use a space before the degree sign, while placing a space in Line 181.

**Author response RC2-37**: Will be fixed in typesetting.

**Author modification RC2-37:** degree sign removed from Kelvin units. Space fixed.

**RC2-38: Line 179**: Here you write "6–7 K".. in Line 181 you write "4.5 K km-1 to 8 K km-1". I would try to be consistent throughout the manuscript and either write "6–7" or "6 to 7".

**Author response RC2-38**: Will be fixed in typesetting.

**Author modification RC2-38:** updated, Line 173

**RC2-39: Line 253**: "reported by Schmidt et al. (2017)". Only the year in parentheses.

**Author response RC2-39**: Will be fixed in typesetting.

**Author modification RC2-39:** Fixed, see Line 245.

**RC2-40: Equation (1) and LIne 208**: Maybe use the common abbreviations HS and HL, or QH and QL (sometimes QE), or SH and LH for the sensible and latent heat flux here and elsewhere in the manuscripts and plots.

**Author response RC2-40**: Currently the naming in the NetCDF files from the model use these variable names. To limit confusion between the manuscript and figures we aim at keeping the variable names consistent.

**Author modification RC2-40:** No changes to the manuscript

**RC2-41: Line 435:** Please remove space after "respectively" and before the comma.

**Author response RC2-41:** done.

**Author modification RC2-41:** Done see Line 432

**RC2-42: Line 513 (Data availability)**: The reference NLSI (2019) is missing in the bibliography.

**Author response RC2–42**: Will be added

**Author modification RC2-42:** Remove the NLSI (2019) reference and referred to the web page.

**Figures and tables:**

**RC2-43:** Most people know what T2, SWin and LWin mean. Nevertheless all tables and figures should be completely readable without the main manuscript. Therefore, it would be good to explain all abbreviations in all captions if they are not explained in a legend within the plot. E.g. T2 SWnet, LWnet SHF, LHF.

**Author response RC2-43**: Will be added and modified accordingly.

**Author modification RC2-43:** Caption text modified in Figure 2, 4, 6 and 7 to explain all abbreviations in all captions.

**RC2-44: Figure 1:** Can you add the latitude and longitude to the axes? The Vatnajökull map is missing the full glacier name. Some of the weather station names are not readable and can only be assigned to Table A1 by excluding the others. A scale in all maps would also be handy.

**Author response RC2-44**: Will be added and modified accordingly. We will update the map with lat and lon. In the text we will add that the scale for the glaciers is the same and the scale for Vatnajökull applies to all the glaciers to save space in the panels.

**Author modification RC2-44:** We added lat/lon to the overview map. In the text we added: *The scale for Vatnajökull (V) applies for all glacier maps (L, H, D, M and V).* The full name of Vatnajökull added to the V-panel. AWS names adjusted for clarity.

**RC2-45: Figure 2**: Please add to the caption a short description of what can be seen in the different rows (RAV2, ICEB and FCST) of the tables. And perhaps a reference to the table with the additional statistics can be added.

**Author response RC2-45**: Will be added.

**Author modification RC2-45: Added to the caption** text: *Different rows indicate different data sources (RAV2, ICEB and FCST) used in the comparison, see Section 3.3.*

And

*Further details are in Table 2.* Referring to the overview data tables for the data.

**RC2-46: Figure 3**: These sentences are redundant: "Note that the colour scale varies between months. Note that the scale varies between panels." The second one would be enough. Is it impossible to see anything in e.g. April if the same scaling is used for all panels? I understand the problem, but it would be extremely helpful using the same scale for all panels visualising the seasonal evolution and to support the statements made in section 5.1.

**Author response RC2-46**: This is a duplicate and will be removed. We tested both fixed and variable color scales. The results were to have them variable to have some insight into the

monthly spatial variability. Figure 7 shown the data on fixed axis per glacier to understand better month to month variations.

**Author modification RC2-46:** Duplicate removed. Colorbar not changed.

**RC2-47: Figure 4**: The vertical scale varies between the panels as well. The label intervals of the x-axis are random. The LWnet panel for Hofsjökull has 5 intervals in a range of 8 W in 2 W steps. The others, for example Drangajökull, have only 2 intervals in the range of 5 W: minimum and maximum for LWnet. I think the different scales of the x-axis are chosen for a reason. If it is possible otherwise, I think this would be preferable. The LWnet gradient for Hofsjökull looks steeper than that of Vatnajökull. This is only due to the scaling of the axis.

**Author response RC2-47**: The axis, both vertical and horizontal, are set to match the data shown in the figures best, per glacier.

**Author modification RC2-47:** see comment above

**RC2-48: Figure 5**: Have the grid points that are in one bin been weighted in any way? Or do some bins only consist of, for example, two grid points and others bins of 300?

**Author response RC2-48**: No weighting has been done. Averages were calculated from all the available grid points, they are not normalized.

**Author modification RC2-48:** see comment above

**RC2-49: Figure 6**: Please add somewhere in the caption "mean monthly" and an explanation of SWnet, LWnet, SHF, and LHF. Furthermore you can try to use thicker lines for melt energy, albedo and cloud cover to increase their visibility.

**Author response RC2-49**: As the caption states "The melt season mean… " is shown here not monthly data. We have tested various combinations of line thickness to find a good balance between bars and lines. This seems to be a good midway.

**Author modification RC2-49:** see comment above. Explanations added for SWnet, LWnet, SHF and LHF.

**RC2-50: Figure 7**: I like the box plots, but an explanation of what we can see would be good. There are different variants of box plots. Which percentiles, mean/median, ….. Furthermore, the caption needs a short statement explaining SWnet, LWnet, SHF, and LHF. Here the information that the scale varies is missing.

**Author response RC2-50**: We will add explanations in the text regarding what quantiles the boxplots are showing. We will add explanations to SWnet, LWnet, SHF and LHF. Scales will be made fixed for all plots at the request of Reviewer 1.

**Author modification RC2-50: Scales are fixed between panels.** Explanations added for SWnet, LWnet, SHF and LHF.

Explanation added for the boxplot:

 *The line inside of each box is the sample median, the top and bottom edges of each box are the upper and lower quantiles (0.25 and 0.75), respectively. The whiskers that extend above and below each box connects the upper/lower quantiles to the nonoutlier maximum/minimum*

**RC2-51: Table 2**: Are all results statically significant? In the 2021 TC paper you explained statically significance using the p value. Maybe you add the p value presentation to the manuscript as well.

**Author response RC2-51**: We will add information about relationship significance.

**Author modification RC2-51:** All the results in the table show a significant relationship tested at 0.05 significance level. We have added this in the table caption.

**RC2-52: Table A1:** I recommend writing the full names of the column headings in the caption. Elevation (Ele.), Number of air temperature in 2 m measurements (N. T2 obs.),...

**Author response RC2-52**: We will write out the full names.

**Author modification RC2-52:** Full names in column headings written out and table heading modified accordingly.

---

## Author Response (AR2)

**RC2: 'Comment on egusphere-2022-1088', Anonymous Referee #2, 21 Jul 2023**

Reviewer comments in **black**
Author modifications in **red – submitted 16.08.2023**

**General comments**

**RC01:** The authors have addressed my previous comments well. Particularly the inclusion of the uncertainties section was a big improvement. I have only a few minor additional suggestions on the manuscript:

**AC01:** We appreciate thorough review of our work. It has improved the manuscript.

**RC02:** L81: change "direct impact glacier runoff" to "direct impact on glacier runoff"

**AC02:** Fixed, see L81

**RC03:** L239: since the authors have added a validation of the outgoing radiation, the section no longer only evaluates the model forcings. I would suggest changing the title of 4.1 to "validation of meteorological forcing and model outputs" or similar

**AC03:** Fixed, see L239

**RC04:** Section 4.1. Although the turbulent fluxes are not measured, could they not still be calculated using the bulk method? In this case, the whole energy balance in the model could be evaluated. If the windspeed is calculated at a different height, a simple logarithmic relationship could be assumed to get the values at 2m. And even if the fluxes cannot be calculated for the whole period, due to data gaps, could it not be used to evaluate part of the period?

**AC04:** Correct. For the work presented here we choose not to do so.

**RC05:** L346-47: Currently, the sentence states: "The bare-ice areas generally reach a certain lower limit (0.1-0.25) limiting further radiative forcing" - surely there is still radiative forcing even if the albedo reaches a minimum value? Do you mean that it limits the effect of the albedo on the radiative forcing?

**AC05:** Yes, this is poorly worded. We have modified the sentence to, see L345:

*"The bare-ice areas generally reach a certain lower limit of albedo (0.1–0.25), limiting further effects of albedo on short-wave radiative forcing, although the timing of bare-ice exposure is important."*

**RC06:** L368: What is meant by higher values being observed at Drangajökull and NE Vatnajökull? Higher values of SW or total energy balance? And is it higher values for all years, or all years excluding 2010 and 2011? Please clarify in the text.

**AC06:** Here a typo was pointing to the wrong figure, see L368

„As shown in Figure 4…" has been changed to „As shown in Figure 6…" and „Higher values were observed for Drangajökull and the northeastern outlet of Vatnajökull" removed.

**RC07:** L503: what is meant by "obvious" trends? An increasing trend of how much?

**AC07:** The sentence refers to trends in warming, increased mass balance etc, pointing out that even if we do not find significant changes over the 20-year period they truly exist in Iceland.

We rephrased the sentence to, see L503:

*"Trends over longer timescales for glacier runoff and increased mass loss of Icelandic glaciers are obvious and have been confirmed in other studies."*

**RC08:** L566: Wittmann et al., 2017 and Schmidt et al., 2017 both use HIRHAM5, which should calculate the cold wave in the snow during the winter. They therefore do not assume a zero sub-surface heat flux, as it will vary over the winter.

**RC08:** Wittmann et al., 2017 and Schmidt et al., 2017 both use HIRHAM for their climate forcing data. For comparison of AWS data (SEB from AWS) they used calculations schemes developed by Guðmundsson et al., 2006. The model does not include a subsurface module.

We modified the sentence from:

*„The assumption of zero sub-surface heat flux has been applied in many recent studies of energy balance and surface melt for Icelandic glaciers"*

To in L566

*„The assumption of zero sub-surface heat flux for AWS data has been applied in many recent studies of energy balance and surface melt for Icelandic glaciers"*

**RC09:** Figure 4: Consider using thicker lines, as they are a bit hard to see (particularly the yellow)

**RC09:** Lines have been made thicker.

**RC10:** Figure 7: consider expanding the y-axis, so that the glacier names do not overlap some of the plotted points.

**RC10:** Text move to the right side on each panel. No overlap.

**RC11:** Figure B3: Mýrdalsjökull is misspelled in the title of both subfigures

**RC11:** Updated and fixed. The figure for Hofsjökull also had a typo, fixed as well